# "A very first clue on the subject": A focus group study on users' perspectives on German plain language summaries of psychological meta-analyses

Claudia Breuer[1,2☯]*, Marlene Stoll[3,4☯], Gesa Benz[3], Mark Jonas[3], Martin Kerwer[3], Anita Chasiotis[3]

**1** Institute for Evidence in Medicine, Medical Center - University of Freiburg / Medical Faculty - University of Freiburg, Germany, **2** Cochrane Germany, Cochrane Germany Foundation, Freiburg, Germany, **3** Leibniz Institute for Psychology (ZPID), Trier, Germany, **4** Leibniz Institute for Resilience Research (LIR), Mainz, Germany

☯ These authors contributed equally to this work.
* breuer@cochrane.de

## Abstract

### Background

Plain Language Summaries (PLSs) are textual summaries of scientific studies that are targeted at lay audiences. PLSs of psychological meta-analyses may benefit laypeople interested in psychological research, psychological practitioners and science communicators. This study explored the perspective of these three user groups on German PLSs of psychological meta-analyses. We wanted to understand the aims and benefits of reading psychological PLSs form their perspective, and how these PLSs should be designed to be useful to them.

### Methods

We conducted four focus group interviews in total. Two focus groups comprised 15 laypeople in sum, one focus group interview comprised six practitioners, and one focus group comprised five science communicators. Interview transcripts were analyzed using content analysis according to Kuckartz and Rädiker. We used main categories for PLS aims and characteristics from a previously developed framework model to classify the content.

### Results

All user groups appreciated the PLSs as a first insight into a broader subject. Additionally, laypeople primarily intended to use the PLSs to gain knowledge, comprehend psychological research and make informed decisions. Science communicators considered the PLSs mainly as a starting point for in-depth investigation of certain

**Data availability statement:** All relevant data are within the paper and its Supporting Information files.

**Funding:** This work was funded by internal funds of the Leibniz Institute for Psychology (ZPID). The authors received no third-party funding for this work. The funders had no role in study design, data collection and analysis, decision to publish, or preparation of the manuscript.

**Competing interests:** The authors have declared that no competing interests exist.

topics, while psychological practitioners regarded them rather as good educational material for clients or patients. Participants across all groups valued plain language, clear structure and short length of PLSs as well as reporting of conflicts of interest related to the presented research. Suggestions for improving the design included a general explanation of technical terms in the form of a glossary, as well as presenting the key message at the beginning.

## Conclusions

The PLSs of psychological meta-analyses investigated in this study have the potential to improve the accessibility of psychological scientific knowledge and provide an attractive first insight into a more extensive subject. The user groups' thoughts and ideas for improvement were implemented in the further development of the PLSs that are published by the Leibniz Institute for Psychology (ZPID).

## Introduction

### Problem formulation

Scientific evidence is relevant for the future of our society and should be accessible to everyone. If successful, science communication to the public can bolster people's scientific literacy, thus helping them to question and verify claims and to engage in public discourse [1]. In a representative German survey in 2021, 54% of respondents from the public reported interest in scientific research, and 40% of respondents stated that they informed themselves about research and science frequently [2]. At the same time, 30% of respondents said that scientists make too little effort to inform the public about their findings [2]. It has been posited by authors such as Kaslow [3] that greater emphasis should be placed on the effective communication of psychological findings. This is due to the growing importance of psychological research findings. Psychological research findings are instrumental in the development of solutions that have the potential to enhance the quality of life for individuals. Also, they play a crucial role in addressing global challenges, such as climate change, that affect society as a whole. The insights derived from psychological research contribute to the development of effective responses to these challenges (e.g., by studying human factors design options for improving the attractiveness to users of climate-friendly choices) [3; 4; 5]. Kaslow argues that psychological science should be communicated to a broad range of individuals and not one single audience, including psychologists who work as practitioners, other professionals, policymakers, and the general public [3]. In a survey conducted in 2020, Genschow et al. [6] found that German psychologists are less engaged in science communication than their international colleagues. The reasons stated for this include a lack of time, low prioritization and doubts about their own competence to translate psychological evidence for the public [6]. Thus, researchers need to find approaches to translate psychological scientific evidence for the public and to guide scientists to do so.

Text-based plain language summaries (PLSs) represent one approach for communicating research findings to the public. There are also other formats using plain language, such as videos or graphics, which are sometimes called Plain Language Resources [7]. The terminology in this field is not yet completely standardized [7; 8]. We use the term PLS for short textual summaries of scientific studies, written in a lay-friendly manner so that lay audiences can easily understand the described research [8; 9]. They may also benefit practitioners and researchers from other disciplines who do not have the time or expertise needed to fully understand the summarized study. PLSs are published across a range of fields and topics, such as planetary science [10], geophysical research [11], autism [12], but most of all in medical sciences [e.g., 9, 13]. One core characteristic of PLSs is the use of clear and simple language to describe scientific studies [8]. This is especially challenging for psychological research as the jargon of psychological researchers often sounds like plain language, but is actually a technical term (e.g., motivation) [14]. Translating psychological findings in a plain manner also implies the need to describe complex methodological issues, such as meta-analytical methods. Because these meta-analyses combine data from multiple studies and therefore provide more robust and reliable results, they are of special interest when it comes to communicating with the public. With the mission to communicate scientific evidence of psychological meta-analyses to the general public in Germany, and to overcome all above-mentioned challenges, the project "PLan Psy" started at the Leibniz Institute for Psychology in Trier, Germany, in 2020. Its aim was to develop an evidence-based guideline for writing PLSs of psychological meta-analyses. For this purpose, various empirical studies were conducted. The leading question for the project was "What should a good PLS of a psychological meta-analysis look like?" One of the cooperation partners of the project was Cochrane Germany, which has many years of experience and expertise with PLSs on medical systematic reviews with meta-analyses.

**What is a good PLS?** The first question to begin such a project with was the general question: "What is a good PLS"? To approach an answer, a literature review was conducted [8]. We found that there is not yet consensus on the ontological and normative criteria for PLSs, i.e., what identifies a PLS as PLS, and what distinguishes a good PLS from a bad PLS [8]. To structure the existing knowledge about PLSs, we developed a conceptual framework model based on scientific literature on PLSs that includes six categories for PLS aims and PLS characteristics each, to which PLS writing criteria and research outcomes can be subordinated [8]. With the help of this conceptual framework model, a first answer can be given to the question of what constitutes a good PLS: its characteristics should be shaped in such a way that the PLS fulfills its prespecified aims [8]. But what are the prespecified aims of psychological PLSs? We needed to identify criteria that are eligible for this discipline. The second question therefore was: "What are the criteria for a good PLS of psychological meta-analyses?". To find an answer to this question, we conducted 4 experimental studies with large, representative samples and derived criteria for a good PLS of psychological meta-analyses from the results [14; 15; 16].

**What is a good PLS of psychological meta-analyses from a reader's point of view?** Up to this point, we have derived the necessity and usefulness of PLSs on psychological topics for the public from scientific, theoretical and societal considerations and we identified the mere absence of such an offer. While the perspective of readers on PLSs on medical and health topics has been explored [17; 18], to our knowledge the perspective of readers of PLSs on psychological topics has not been explored in detail. As we have argued previously in the context of experimental studies on psychological PLSs [14], we propose that a separate investigation of psychological PLSs is essential for the following reasons: first, psychological research is distinct in that the subject is to examine the mind. So, the constructs of interest are often not directly observable, but must be inferred or operationalized by manipulating proxy variables. Psychological studies often rely on correlation designs and context-sensitive interpretations. Thus, psychology is a very nuanced and complex field with a growing complexity of research methods (see also [19]). Finally, the technical terms used as psychological jargon often overlap with plain language terms used in everyday language: For example, the prominent Big Five Model of personality was developed using a psycholexical approach under the rationale that individual differences that people consider to be important are reflected in everyday language [20]. Therefore, the Big Five Model of personality derives its nomenclature from everyday language, which can be seen in

the naming of the five personality dimensions: openness, conscientiousness, extraversion, agreeableness and neuroticism are words that are (more or less) used in everyday language, too. In this example, the scientific term, e.g., "agreeableness" or "agreeable", sounds like everyday language but is a technical term. Furthermore, a recent phenomenon that has come to light is the overuse of psychological technical terms in everyday language, which results in the expansion of their meaning: Concept Creep [21]. For example, the psychological term "trauma" was defined as a psychological disorder caused by an extraordinary event, for example in wartime deployment. Over the years, the term "trauma" or "traumatized" is used by more and more people to describe reactions towards comparingly less grave events as for example marital infidelity [21].

In the experiment mentioned above [15], we explored participants' responses when asked to formulate their impressions of the PLSs: What did they like? What did they dislike? The evaluation of the answers of these questions can be found in Stoll et al. [22]. One result of this evaluation was that the readers seemed to have different needs in terms of text length and information content. Some found the PLS too long, some too short, some deemed it exactly right. The question arose as to whether the difference in prior knowledge or reading objectives when reading the PLS could be the cause.

In order to gain deeper insights into different users' perspectives and their aims, needs and experiences when reading PLSs on psychological meta-analyses, the present qualitative study was designed. The target group of psychological science communication is heterogeneous, since many individuals benefit from psychological evidence – psychologists, other professionals, policymakers, students, and the general public [3]. To narrow down the target group for which the PLSs are written, we assume that only people interested in psychological evidence will read the PLSs. Of those, we identified three subgroups that may have different prior knowledge on psychology: 1) laypeople interested in psychological evidence, 2) practitioners in applied fields of psychology, and 3) science communicators (i.e., individuals communicating science to the public such as science journalists). Since the professional objectives of these user groups vary, it can be assumed that their aims of reading PLSs will differ and that they may benefit from the PLSs in different ways. So finally, we are interested in identifying these groups' common or differing aims when reading PLSs and their common or differing ideas on how a PLS should be designed in order to be useful.

Qualitative studies on PLSs or other knowledge translation tools have been done before. Barbara et al. [23] explored citizens' perceptions of the usability of information on a web portal on evidence-informed healthy aging. Ellen et al. [18] conducted interviews with managers and policy-makers to obtain feedback about user-friendly summaries of systematic reviews. Anzinger et al. [24] used a mixed method approach to generate guidelines to improve knowledge translation tools that make child-health evidence understandable for parents. Glenton et al. [17] developed different versions of PLSs and tested them among members of the public using semi-structured interviews. They used the results to revise the PLSs and produce new PLS versions, which were later re-tested.

To gain insights on users' viewpoints and to further develop the writing guideline for PLSs of psychological meta-analyses, we therefore conducted a qualitative focus group study in addition to the quantitative studies already conducted. This study was planned, conducted and analyzed as a cooperation project between the Leibniz Institute for Psychology, the Institute for Evidence in Medicine at the University of Freiburg, and the Cochrane Germany Foundation.

## Research question

This focus group study aims to examine the perspective of three user groups of PLSs on psychological meta-analyses – laypeople who are interested in psychological science, science communicators, and psychological practitioners. Our research questions are: What are the aims and benefits of reading PLSs from the different user groups' (interested laypeople, practitioners, and science communicators) point of view? How should PLSs, from the users' point of view, be designed in order to be useful?

## Methods

We report this study in accordance with the consolidated criteria for reporting qualitative research [COREQ; 25] (see S1 Table) and the Guidance for Reporting Involvement of Patients and the Public [GRIPP2; 26] short form reporting checklist (see S2 Table). The study was approved by the ethics committee of the University of Trier, Germany (No. 47/2021) and preregistered [27].

### Study design

This was a qualitative study using online focus group interviews. Focus group interviews are group interviews in which the reactions of a group of participants to particular information as a collective orientation is investigated [28]. As it is the aim to supply attractive and, above all, valuable PLSs, we used this approach to involve the perspectives of potential user groups in the improvement of the writing guideline for PLSs of psychological meta-analyses.

We aimed to conduct two focus group interviews with laypeople, one focus group interview with science communicators, and one with psychologists working as counselor and/or psychotherapists ("psychological practitioners"). To approximate the recommended sample size of six to ten participants [28], we invited up to nine participants per group. Achieving data saturation was not an explicit objective of the study.

During the focus group interviews, the participants received three PLSs (see S1 File). All PLSs were written in German based on the criteria of the first version of the PLS guideline [29] by AC, MK, MS and GB, who are employees of the Leibniz Institute for Psychology and who have experience with writing this kind of PLS. The guideline provided specifications for the structure and content characteristics and included specific text modules. This ensured that all PLS were created in the same way. The specifications were organized into five categories: linguistic characteristics, formal attributes, general content, presentation of results, and presentation of the quality of evidence. Linguistic characteristics, for example, referred to the replacement of technical terms with language suitable for the target audience. Formal attributes specified structural elements such as a maximum text length. General content referred to what the PLS should include, for example, a title related to the key message of the text. Presentation of results focused on providing in addition to quantitative effect sizes, a qualitative explanation of the observed effects. Presentation of the quality of evidence, for example, required a paragraph discussing potential publication bias, fostering transparency regarding the reliability of the presented results.

### Setting and participants

Recruitment took place from February 9, 2022, to August 9, 2022. We used a purposive sampling approach. Participants were recruited through various channels. Laypeople were reached via the website of a popular science magazine ("Psychologie Heute"), a Facebook group for study participants, and ads on an online platform ("eBay Kleinanzeigen"). Science communicators were approached through the website of Psychologie Heute, Cochrane Germany's website and Twitter account, the website of a science journalists' association and editorial offices of popular science magazines. Psychological practitioners were recruited via Psychologie Heute, Cochrane Germany's website and Twitter, the website of a professional association and psychotherapy training institutes. For science communicators and psychological practitioners, we additionally used authors' personal and professional networks, because we faced challenges while recruiting participants from these groups. After voluntarily expressing their interest to participate via e-mail, potential participants received an informational letter on the procedure and conditions, and a consent form along with a short questionnaire on inclusion criteria and sociodemographic characteristics (age, gender, education level).

Inclusion criteria were:

• native-level German language proficiency (self-reported)

• legal age (≥ 18 years)

- interest in psychology

    Additionally, depending on the target group:

- laypeople: no degree in psychology

- psychological practitioners: a master's or diploma degree in psychology and working as psychological counselors and/or psychological therapists

- science communicators: professional involvement in science communication including psychological topics

No other characteristics were collected that could indicate any attitudes toward PLSs. For laypeople, the number of submitted consent forms exceeded the number of available spots. While we did not employ stratified sampling or use a recruitment panel, we aimed to optimise variation in socio-demographic characteristics by purposively selecting individuals to achieve a heterogeneous sample. Therefore, we included the few individuals with lower education levels and ensured a balanced gender distribution as well as representation of all age decades. For psychological practitioners or science communicators, the number of submitted consent forms did not exceed the number of available spots.

We invited 16 laypeople to two focus groups (focus group 1 and 2), five science communicators (focus group 3) and seven psychological practitioners (focus group 4).

All participants gave written informed consent prior to the focus group interviews and received reimbursement of €30 afterwards. The online interviews were conducted using the video conference software GotoMeeting and were digitally audio recorded. Due to a damaged audio recording of one focus group interview with psychological practitioners ($n = 5$), data from this group could not be analyzed, so the interview was repeated with six newly recruited participants. The four focus group interviews analysed lasted between 94 and 117 minutes (mean: 106 minutes).

## Procedure

The focus group interviews were conducted in German by one of the first authors (CB), who had completed a moderator training for qualitative research before. Apart from the moderator, no other non-participants joined the sessions. All interviews followed a focus group protocol that included general instructions for the participants and a list of questions according to a semi-structured interview guide. The interview guide (see S2 File) was developed beforehand and pilot-tested with non-academic staff members. The procedure was the same in all focus group interviews. They began with a brief introduction of the moderating person, a female researcher in evidence-based medicine. She holds a degree in pharmacy and drug research, and she has not been involved in the writing of the PLSs. Next, participants received information on the study procedure and audio recording. The interview started with some warm-up questions on favorite psychological topics and sources of information used. Afterwards, the participants read three PLSs for themselves for the next 15–25 minutes. The topics of these PLSs were 1) video game impact on perceptual, attentional, and cognitive skills, 2) comparative efficacy of psychotherapeutic interventions for patients with depression and 3) coping as a mediator between locus of control, competence beliefs, and mental health. Then, participants were asked various questions about their first impression of the PLSs, their aims and possible benefits when reading them and their opinion on how PLSs should be written in order to be useful for the user group they represent. Each session ended with a question about willingness to recommend this kind of PLS. Field notes were taken during the session.

## Data analysis

Audio recordings of focus group interviews were transcribed verbatim. They were not returned to participants for commenting. To describe the data in a methodologically controlled, transparent and computer-assisted way, we followed the structuring content analysis approach according to Kuckartz and Rädiker, using a deductive-inductive categorization

 

process [30]. Based on initiating text work, two researchers (CB, MS) independently applied main categories from Stoll et al. [8], who had already identified PLS aims and characteristics through a systematic literature review. Additional inductive main categories were added if they could be derived from the data. Codes were continuously compared, and discrepancies were solved by discussion. In a second step, one researcher (CB) developed inductive subcategories within main categories and applied these subcategories to the data. Unclear cases were solved by discussion with MS. Results were presented to and discussed with co-authors. The data was managed using MAXQDA software. Participants were not involved in discussing the findings. During the focus group interviews, science communicators and psychological practitioners sometimes refrained from expressing their personal needs and instead talked about what they believed would be effective for a lay audience. When this occurred, the moderator redirected the conversation to focus on their own professional perspectives. We excluded such statements from our analysis, as our research aim did not include investigating, e.g., science communicators' assumptions of laypeople's needs. All answers were given in German but were translated to English for this publication. The authors used ChatGPT in order to improve the translation of participants' quotations.

## Results

### Participants' characteristics

Of the 28 invited participants, two did not take part, with one layperson due to unknown reasons and one psychological practitioner due to technical problems. Participants in the focus groups were predominantly of younger age (mean: 35.6 years). Only two laypeople had a lower secondary education. The gender in the focus groups was balanced except for the practitioner group, which had only one man and five women. Participants' sociodemographic characteristics are shown in Table 1.

### Users' aims

We categorized six deductive main categories for aims and expectations according to Stoll et al. [8] across all focus groups. These categories were 1) accessibility, 2) understanding, 3) knowledge, 4) empowerment, 5) communication of

Table 1. Characteristics of participants (N = 26).

|  | laypeople | science communicators | psychological practitioners |
|---|---|---|---|
| Sample | 2 focus groups | 1 focus group | 1 focus group |
| Period | February 2022 | May 2022 | Aug 2022 |
| Number of participants | 15 | 5 | 6 |
| Gender |  |  |  |
| *Female* | 8 | 3 | 5 |
| *Male* | 7 | 2 | 1 |
| Age |  |  |  |
| *18-39* | 10 | 2 | 6 |
| *40-49* | 2 | 1 | 0 |
| *50-59* | 2 | 2 | 0 |
| *60-69* | 1 | 0 | 0 |
| *≥ 70* | 0 | 0 | 0 |
| Highest education level |  |  |  |
| *Lower secondary education (according to ISCED)* | 2 | – | – |
| *Upper secondary education (according to ISCED)* | 8 | – | – |
| *University degree* | 5 | 5 | 6 |

ISCED: International Standard Classification of Education [31].

research and 6) improvement of research. The first four categories depend on each other: Accessible information is neccessary for understanding, which can lead to an increase in knowledge and, thus, allow for informed decisions (empowerment category). No additional inductive main categories were derived from the data. Inductive subcategories are mentioned in Table 2.

**Accessibility.** Many lay participants perceived the PLSs as a quick introduction or short overview of certain psychological topics. Being a short summary of multiple studies, PLSs were seen as time-saving.

"For me, a very first clue on the subject, so that I can engage with it and read more about it in the media or maybe look for some literature on it." (1: 87)

"So of course you might not understand the whole background, but you definitely get a first impression of how these studies turned out and what the results were and just to get an initial overview of certain topics without having to spend ages on it." (2: 139)

Some of them found the PLSs helpful as a kind of "preview" for deciding whether the topic interests them enough to dive deeper and search for additional information afterwards:

**Table 2. Users' aims and benefits of reading PLSs, according to categories from Stoll et al. [8].**

| laypeople | science communicators | psychological practitioners |
|---|---|---|
| **Accessibility**: low-threshold information on research that is easy to find, highly visible, freely accessible, attractive, appealing, and technically accessible. | | |
| - topic introduction/overview<br>- summary of multiple studies<br>- reduce inhibitions towards stigmatized psychological topics | - search for topics to report about<br>- starting point for research<br>- overview of meta-analysis | - summary of research papers<br>- avoidance of extensive searches for meta-analyses<br>- access to research outside their own field of expertise |
| **Understanding**: information on research (including research questions, methods and results) that is understandable. | | |
| - understanding of research methods<br>- explanation of technical terms | - reading original paper/contact to meta-analysis authors for a better understanding of the results | - understandable explanations for consultation or therapy sessions |
| **Knowledge**: increasing knowledge about specific subjects based on scientific evidence. | | |
| - knowledge acquisition<br>- underpin own opinion | – | - knowledge acquisition for clients/patients |
| **Empowerment**: to make informed, self-determined decisions, and to foster public participation in decision processes. | | |
| - help with decision-making<br>- everyday support<br>- positive behavior change<br>- preparation for sessions with psychologists/physicians | – | – |
| **Communication of research**: enhancing the communication and dissemination of research by addressing a broad audience. Thereby, the trust in and impact of science on daily decisions as well as on political decisions and actions is thought to increase. | | |
| - scientifically grounded findings<br><br>- explanation of scientific methodology<br><br>- transparent reporting<br><br>- credibility<br><br>- entertainment | - overview of topics and trends | – |
| **Improvement of research**: making a contribution to the improvement of research practice itself and of interdisciplinary communication. | | |
| – | – | - stimulate reporting of precise definitions and summarized reporting of research |

"It's also for a general overview. For example, when I want to watch a movie or a documentary, I usually inquire about the topic beforehand or deliberately choose a subject that interests me. After reading the text, I know whether it's worth delving deeper into the topic for me." (1:93)

The texts were even suggested to help reduce perceived inhibitions towards stigmatized psychological topics like depression, encouraging people to engage with such subjects.

Some lay participants found that direct links to additional information sources were lacking.

Science communicators suggested the PLSs in general as a helpful initial source to search for psychological topics, to start their research on specific topics and to gain an overview of a meta-analysis:

" Well, I don't know how they will finally be presented on the website, but if you can immediately see what it's about with just one click, I would also pick out possible topics there." (3: 101)

Some psychological practitioners found the PLSs to be a helpful short, time-saving summary of research papers. One participant recognized that a collection of PLSs on a web platform replaces extensive searches for original meta-analyses. Others saw no advantage compared to scientific abstracts but found the PLSs helpful to access research outside their subject area:

"But I could imagine, if it was more in the pedagogical or sociological context and maybe I'm interested in a topic where someone had asked me something about, that I might not find a bad entry point there." (4:89)

**Understanding.** For laypeople, the PLSs were a time-saving instrument to understand methods and content of scientific research, so the comprehensibility of PLSs was very important for them:

"And that they are also clearly understandable for me and not in technical jargon and that I still have to run to my doctor with the texts and say: "Can you translate what this means for me?" (2:183)

They aimed to comprehend the complete texts, so they wanted explanations for all technical terms and they mentioned difficulties to interpret the different effect sizes in all texts:

"For example this effect size, what does that actually mean? What does it tell you?" (1: 52)

Some of the science communicators said depending on the importance of the study they would read the original paper or even contact the study author or the press contact for a better understanding of the results and their meaning:

"But usually, there are always new pieces of information that come out from a conversation with researchers, things that you didn't have in your mind before, I think. Also, assessments, for example. How important is this particular result, what does it mean? That becomes more apparent when you actually talk to people, because it's no longer as cryptic as it is in the paper." (3: 114)

Psychological practitioners saw the PLSs as a helpful source to find understandable explanations they can use in their therapy or counseling sessions.

**Knowledge.** Many of the lay participants stated they would use this kind of PLS to gain knowledge or underpin their point of view:

"Yes, also I definitely would say [I would read the text] to learn new things." (2: 139)

Some of them mentioned especially the key message of the PLSs to be easy to remember.

However, others said they would not read such PLSs in the future because the texts do not contain important or new information:

"To be honest, these texts we've read now wouldn't help me at all, because they have so little content that I wouldn't build on that in any conversation, discussion whatsoever." (2: 185)

In contrast, psychological practitioners suggested PLSs as valuable information material for their clients or patients to gain knowledge:

"And I would definitely hope that they can gain additional knowledge for themselves, beyond what is covered in the therapy session." (4: 81)

**Empowerment.** Many lay participants perceived the PLSs as helpful for decision-making, positive behavior change or as support in everyday life:

"[...] where you are at a crossroads, not knowing whether to go left or right, and this can provide guidance." (1:89)

"Or it then also has an impact on one's own behavior and thinking." (1:123)

"Maybe it will somehow help me in life or something. And then it's interesting to have read something like that." (2:137)

Particularly the presented PLS on psychotherapeutic interventions for patients with depression was seen as possibly helpful:

"[...] for example, with the psychotherapies, [it] could be an initial impetus for people who are affected to say: "Ah okay, it is proven to somehow help". Maybe you can motivate people with something like that." (2: 141)

However, one lay participant stated that "[...] the topics, at least as they are presented, don't offer much in terms of integrating them into everyday life." (1: 50)

Laypeople also suggested to use the PLSs for preparation of appointments with physicians or psychologists:

"For example during psychotherapy, if you can inform yourself beforehand [...]. And then you can go into a conversation with a psychologist and you are not completely unprepared, but able to ask questions, for example." (1: 134)

**Communication of research.** In general, lay participants appreciated the PLSs as an understandable source of information reporting scientifically grounded findings. Some of them explicitly valued the explanation of scientific methodology:

"I think it's good that way and I enjoyed it. It clarified for me how you work on scientific projects in psychology, and I found it understandable, really." (2: 105)

The reporting of possible bias, funding and conflicts of interest was particularly described as transparent and scientific.

Due to their scientific character and Leibniz Institute for Psychology as publisher, laypeople assessed the PLSs as credible information source:

"But that you can definitely be pretty sure that it's not just some blogger writing what he thought or something like that, but that you know you can rely on it to some extent more than on other websites" (2: 177)

One lay participant said he would also read PLSs for entertainment:

"Perhaps alternatively, like watching a movie in the evening or just reading the news, reading two or three articles just for fun and for self-improvement." (1: 187)

For science communicators, a web platform with this kind of PLS was perceived as a helpful source to gain an overview of topics and trends in psychological research.

**Improvement of research.** Improvement of research was an aspect which hardly mattered in all four focus group interviews. Solely one psychological practitioner wanted to recommend the PLSs for inspiration to colleagues working in research:

"Because maybe, especially with technical terms that kind of define and sharpen and summarize things. Maybe it's not a bad thing for the profession as a whole if you had to define your terms better." (4:172)

### Users' requirements for characteristics

In addition to the aims, we categorized six deductive main categories for PLS characteristics according to Stoll et al. [8]. across all user groups. Participants mentioned aspects of 1) linguistic attributes, 2) formal attributes, 3) general content, 4) presentation of results, 5) presentation of quality of evidence and 6) contextual attributes. No additional inductive main categories were derived from the data. Inductive subcategories are presented in Table 3.

**Linguistic attributes.** In general, participants described the language in the PLSs as "easy to read", "concise" and "understandable" and liked the avoidance of "nested sentences". One psychological practitioner explicitly appreciated that the texts were written in German:

"I would read it simply because it's in German and not in English." (4:83)

**Table 3. Requirements for characteristics of plain language summaries on meta-analyses, according to categories from Stoll et al. [8].**

| | Laypeople | Science communicators | Psychological practitioners |
|---|---|---|---|
| **Linguistic attributes**: tone or style of the language, choice of words or text difficulty. | | | |
| plain language | + | + | + |
| native language | – | – | + |
| explanations for all technical terms | + | + | + |
| more personal and informal language | + | + | – |
| avoidance of redundancy | + | – | + |
| **Formal attributes**: characteristics on the formal level such as word limits, structure, graphs and tables. | | | |
| clear structure | + | + | + |
| standard highlighted subheadings | + | + | + |
| short length | + | + | + |
| standard layout | + | – | + |
| important paragraphs highlighted | + | – | – |
| graphics | + | + | – |
| **General content**: PLS' content and the alignment of contents. | | | |
| more detailed description of methodology of included studies and meta-analysis | + | + | + |
| importance of paragraph "key message" | + | + | + |
| importance of paragraph "What do the results mean in everyday life?" | + | + | – |
| results or key message at the beginning | + | + | – |
| abbreviation of repetitive content | + | + | + |
| **Presentation of results**: e.g., handling of statistical terms and numerical data. | | | |
| explanation of effect sizes | + | + | – |
| graphics | + | + | – |
| balanced presentation of benefits and risks | + | + | – |
| **Presentation of quality of evidence**: e.g., reporting of study design or authors' conflicts of interest. | | | |
| transparent reporting of funding and conflicts of interest | + | + | + |
| reporting of publication bias | + | + | – |
| statement on reliability of results | + | + | + |
| reporting of study design with information on causality or correlation | – | + | – |
| quality of included studies | – | – | + |
| clarification if interpretation done by meta-analysis authors or PLS authors | – | – | + |
| **Contextual attributes**: general context of the PLS. | | | |
| clarification of target audience | + | + | + |
| information for professionals in attachment | – | + | – |
| clear presentation of authorship | + | – | + |
| barrier-free | + | – | – |
| access without charge | + | – | + |
| linkage to included studies and original paper | + | – | – |
| highlight PLSs of important, new meta-analyses | – | + | – |
| citation of original paper | – | + | – |
| pdf-download, print, forward reference | – | + | – |

*(Continued)*

**Table 3.** (Continued)

| | Laypeople | Science communicators | Psychological practitioners |
|---|---|---|---|
| press contact | – | + | – |
| search engine optimization | – | + | – |
| retrievable via search in common scientific databases | – | – | + |
| attached to original paper | – | – | + |

+: aspect mentioned, -: aspect not mentioned

Nevertheless, several technical terms (e.g., different types of psychotherapies, "perception", "attention", "control for negative events") mentioned in the texts were not well understood, so participants in all groups wished for more explanation, for instance in the form of a glossary, text boxes, mouseover or in brackets next to the term. Two participants from each of the laypeople and science communicator groups stated that they would prefer a more personal or informal language. Several sentences in the texts were perceived as redundant by one layperson and one psychological practitioner.

**Formal attributes.** Participants of all groups praised the PLSs' clear structure. The standard bold subheadings phrased as questions across all PLSs were perceived as very helpful for orientation in the text, searching for certain aspects and skipping of redundant paragraphs if required. Participants appreciated the PLSs' short length, nevertheless, some lay participants felt the text was "bloated" (not much content for a long text). Laypeople and psychological practitioners perceived the standard layout well in terms of recognition value, but some laypeople wanted important paragraphs to be highlighted. Lay participants and science communicators suggested integrating graphics into the text, for instance, charts to explain numerical data, "[...] because that also immediately makes you more inclined to take a look at the text." (2:237) and "[...] if I understand it right away, I can also explain it more easily to a layperson." (3: 138) Moreover, science communicators found graphics helpful to incorporate them in their own work, in a video for example. Conversely, one lay participant explicitly stated that he "[...] wasn't bothered by the fact that the text was just a continuous piece of text [...]" (2:71).

**General content.** For many participants in all groups, content of the PLSs was not sufficient. They called for more information on nationality of included participants, investigated interventions, measurement methods, reference to and quality of included studies, search strategies, statistical analysis and researchers. Laypeople stressed that lacking information made it difficult for them to comprehend the research:

"So that I can decide for myself whether it makes sense to compare these studies. Can anything meaningful come out of these studies at all?" (2:75)

Nevertheless, one lay participant stated "It was so and so many people in the year so and so, so and so examined. As a layperson, you just don't remember that. I think most people are not necessarily interested in whether it's the year 2014 or 2016 and how many, but rather the main message." (2:65)

Science communicators perceived the PLSs as a helpful "initial overview [...] maybe just to see if the topic interests me and what the key findings were." (3:77) and wanted to gain lacking information from the original paper or the meta-analysis authors directly. For most psychological practitioners, the PLSs were not a source of information for themselves. They preferred to use the more detailed abstract or the original paper. However, participants from all user groups appreciated the paragraph "key message", while laypeople and science communicators also explicitly appreciated "What do the results mean in everyday life?". Laypeople and science communicators suggested changing the sequence of paragraphs. Many laypeople and science communicators wanted to read the results or the key message at the beginning of the text.

"But first I want to know what actually came out of it, because most of the time I'm searching because I want a result and not because I want to know exactly how the study was designed." (2:207)

Science communicators found the structure in general too scientific and suggested aligning the content "from important to detail" or provide a lay friendly shorter version with an additional detailed text for science communicators afterwards.

The explanation of meta-analysis at the beginning of every PLS was perceived as important content in all groups, but as too long and redundant when several PLSs are read. A small box with a definition next to the text, color highlighting or the possibility to skip the definition was suggested as an improvement.

**Presentation of results.** The unexplained effect sizes were the most criticized aspect by laypeople and science communicators. Most participants were not familiar with them and were confused by the different effect measures mentioned in the texts. They asked for more information on the scale, the effect calculation and asked for a definition of the narrative description of results (e.g., meaning of a "moderate difference"). Graphics were suggested as a helpful tool to explain or replace numerical data. Nevertheless, some science communicators explicitly advocated for reporting of numerical data in the PLSs for the sake of traceability. Psychological practitioners indicated that they understand the various effect sizes. Some laypeople and science communicators perceived the reporting of conclusions of the PLS on video games to be too positive. They suggested a more balanced presentation of benefits and risks, including aspects beyond the scope of the underlying meta-analysis.

"Actually, only positive things are being said here. It is stated that playing action video games has a positive impact or correlates positively with various measures of cognitive abilities. However, in my opinion, the entire discussion around action video games is not primarily about whether playing them makes you smarter or dumber in terms of cognitive functions like perception speed. Rather, it's often about whether they glorify violence or potentially lower the threshold for violence among young people [...] but perhaps it would be a good idea, especially for topics that have a larger societal debate, to at least acknowledge this by saying, maybe in just one sentence, 'Of course, this is not the only aspect of interest, but here in this paper, we are focusing solely on this one aspect. However, there are the following aspects of the debate about topic XY that must also be considered." (3:116)

**Presentation of quality of evidence.** Many participants appreciated the transparent reporting of funding, conflicts of interest and publication bias:

"That reinforces, for me, this, let's say, scientific character. You can also see that it's not some kind of lobbying." (1:71).

Nevertheless, some laypeople reported disappointment or mistrust or missed clear conclusions when a potential bias limited the results:

"[...] that I don't believe in the whole thing. And that I wouldn't use this medium [the texts] [...] for other things anymore." (1:67)

"[...] and especially with the statement that more studies are needed. Then I don't take anything away for myself because I think, great, I can't rely on that at all" (2:169)

One lay participant found the reporting of conflicts of interest in the PLS on video games to be unclear:

"[...] that one researcher is a member of this scientific advisory board. A biotechnology company [...]. So I didn't quite understand why that is a conflict of interest." (2:131)

Participants of all groups pointed out that a general statement or ranking about reliability of results, which would help them to assess the meaningfulness and trustworthiness of the results, was lacking:

"Something comes to mind that might help me overall. It's just a kind of basic evaluation. (...) Are there reliable effects or not?"(2:247)

One science communicator stated that a presentation of the study design with an explanation of causality and correlation was missing. One psychological practitioner missed a statement on the quality of studies included in the meta-analysis. Moreover, one psychological practitioner asked for clarification if the interpretations of results were done by the authors of the meta-analysis or by the authors of the PLS.

**Contextual attributes.** Participants of all user groups were unsure about being the target audience of the PLSs themselves.

"[...] to whom is the text actually addressed? On the one hand, it's very simple and easy to read, but then in each of these three texts, there are these two short paragraphs where there is this reference to statistical metrics that I can only understand if I'm familiar with the subject [...]" (1:98)

Many participants found it difficult to present just one text for different user groups, e.g., laypeople and professionals.

"Because, of course, the information is then great for science communicators, but potentially not for laypeople [...] I think you have to decide for whom you would like to publish something like this." (3:87)

It was recommended to define the target group in a short introduction to the PLSs, develop different formats for different groups or to use a lay format and attach additional information for professionals in a separate document.

Some participants suggested a more explicit naming of the PLS author to increase the trustworthiness of the information source.

"I think it's important to know where the whole thing comes from. Honestly if I know it's posted on Wikipedia, it could be from anybody. It's not a reliable source." (2:211).

A barrier-free access without additional paywall was also mentioned as important:

"Because if you say that it should be accessible to everyone, then it really has to be accessible to everyone, even if you're disabled in some way." (1:273)

For better traceability, many laypeople wanted to know which studies were included in the meta-analysis, with direct links in the text or as short summaries in a separate file:

"So, what kind of studies are these? I'm missing citations so that, as a layperson, I have the opportunity to verify it." (1:77) A direct link to the original meta-analysis was also considered helpful to gain further information if needed.

Science communicators were particularly interested in PLSs of newly published, important meta-analyses. Suggested methods to highlight these were to mention the year of publication in the title, options to sort the PLSs by year or being informed when PLSs of new meta-analyses are published. Science communicators missed the full citation of the original paper and opportunities for pdf-Download, print and forward the PLSs to colleagues. Moreover, they further wanted a contact for press enquiries on the meta-analysis. Search engine optimization was considered as important to facilitate finding the PLSs with search engines such as Google. In the psychological practitioners group, finding the PLSs via scientific literature databases and as attachment to the original paper was suggested as helpful.

## Discussion

### Main findings

This study aimed to investigate the user perspective on German PLSs of psychological meta-analyses by conducting focus group interviews with three subgroups of users: interested laypeople, science communicators and psychological practitioners. The first research question concerned the users' aims and benefits of reading PLSs. Many participants from all user groups saw PLSs as a first introduction to a more extensive subject. A further benefit mentioned in all user groups was that reading PLSs would save resources. Laypeople valued that the PLSs provided access to scientific knowledge. They also liked to learn more about how psychological research is conducted and saw the PLSs as decision aid or preparation for sessions with psychologists or physicians. Science communicators perceived the usefulness of the PLSs less in the content conveyed but more in the fact that PLSs provide a quick overview of current topics in psychological research, from which they would delve deeper and contact the authors of the original studies, for example. Psychological practitioners appreciated the summarizing character of PLSs, but many of them preferred reading the abstract or the original paper. Nevertheless, they considered PLSs as good educational material for clients or patients, but also to gain insights into research outside their own field of expertise.

The second research question of our study was how to design PLSs from the users' perspective to ensure usefulness. The participants valued the plain language and clear structure of the texts, but wanted to see more information on technical terms. Further, they liked the transparent reporting of funding and conflicts of interest but missed statements on the quality of included studies. Laypeople and most science communicators found the presentation of effect sizes in the results report incomprehensible. Laypeople and science communicators also proposed to report the key message at the

beginning of the PLSs. Science communicators mentioned aspects that could improve the use of PLSs as a starting point for a deeper investigation. For instance, they suggested providing a press contact.

## Main findings in context

Many of the topics that our participants mentioned in relation to characteristics of PLSs are in line with prior research findings. In terms of linguistic attributes, the issue of understandable language is mentioned by most participants in our study as well as for example in Anzinger et al. [24] and Barbara et al. [23]. Translating scientific findings for the public requires balancing the use of scientific jargon while maintaining understandability and scientific rigor. While our findings from the prior experimental study were inconclusive about the benefits of a glossary when compared to simply replacing technical terms with non-technical terms within the text [14], participants in the current study clearly expressed the desire for a glossary, text boxes or some other form of more detailed explanation of technical terms. This is in line with findings of Barbara et al. [23], who outlined that participants especially liked a glossary in addition to evidence summaries, as well as digital pop-up boxes that appeared directly in the text if one hovered over a term in the body of the evidence summary [23].

Formal attributes like a consistent layout and clear structure with subheadings were also named by participants in the interview study on user experience of evidence summaries and blog posts by Barbara et al. [23]. Ellen and colleagues [18] found that their participants, health system managers and policy makers who read summaries of systematic reviews, wanted the key message to be presented at the beginning of the text, and further information afterwards. This was also stressed by our participants.

Another issue is the amount of details PLSs should entail. In a prior experimental study, we found that providing information on the operationalization of the research question in the PLSs (i.e., how were the theoretical concepts of the research question measured in the summarized studies?) resulted in readers perceiving the PLS as less accessible, understandable and empowering than without this information [14]. Anzinger et al. [24] found that parents who, among other texts, read PLSs that synthesize child-health evidence, were not interested in study characteristics, so the recommendation of Anzinger et al. for PLS writers is to "Be concise, but link to more information. More details confuse parents and reduce readership." [24, Table 4, p.7]. In a previous study, we found that PLS readers had very heterogeneous opinions regarding the optimum length of PLSs [22]. To address this challenge, it may be beneficial to enhance the alignment between the target group's needs and the PLS. For example, as also mentioned by Anzinger et al. [24], to not describe additional information in the text, but offer it in the form of pop-up windows or links. The additional information in this manner does not interrupt the reading flow and can be accessed by interested readers if needed (see also [22]).

While the subject of this study was PLSs on psychological studies, the subject of the majority of previous research on the user experience of PLSs has been PLSs on medical studies. Our study indicates that users' needs in terms of the design of the PLSs are similar between psychological and medical PLSs as determined in previous research. The extent to which there are specific differences in the aims, use and expectations of psychological versus medical PLSs was not in the scope of this study and needs to be investigated in future studies.

The findings of our study were mainly used to further develop a writing guideline for PLSs of psychological meta-analyses. The results of the present study were discussed in the project team that was responsible for developing "KLARpsy" [32], the online platform that provides psychological PLSs written in accordance with the developed guideline. User comments were discussed and incorporated into the writing guideline if it was feasible, for example:

• We changed the content structure and put the key message at the beginning of the PLS, and a standardized paragraph with background information on meta-analyses at the end of the PLS.

• We added a paragraph on the role of the PLS-authors as opposed to the authors of the meta-analyses, to clarify who wrote the PLS and who conducted the meta-analysis.

- We added explanations of various kinds of effect sizes in the corresponding German plain language dictionary "KLARpsy Glossary" [33]. PLSs published on the online platform show these explanations in pop-up boxes andl link to the glossary so that the readers can look up explanations for words that are incomprehensible to them.

Overall, the KLARpsy approach is based on Cochrane's idea of a PLS library for meta-analytical evidence. Its development is evidence-based and all PLSs are written based on an evidence-based guideline [34]. While Cochrane aims to provide PLSs for all systematic reviews that they have produced by themselves and that deal with questions of medicine and health, KLARpsy aims to provide PLSs on systematic reviews with meta-analyses that deal with questions of psychology. KLARpsy is set in Germany at the Leibniz Institute for Psychology, a publicly funded research infrastructure organization that supports the entire scientific working process including publication of studies and science communication. KLARpsy, just like Cochrane, engages with its audience. For example, KLARpsy endeavours to have every text test-read by a non-expert and there are various opportunities for users to give feedback or request information. Unlike Cochrane PLSs, KLARpsy PLSs are only written in German language and now accompanied by a glossary that provides plain language explanations for technical terms in psychology.

## Strengths and limitations

To our knowledge, this was the first focus group study investigating user perspectives on PLSs of psychological meta-analyses. This enabled us to directly involve potential users in the development of the online platform "KLARpsy". By interviewing three different groups of PLS users with varying levels of psychological expertise, we were able to identify differences in terms of potential aims and benefits when reading PLSs. Embedded in the research project "PLan Psy" and the associated theoretical and quantitative research efforts, the insights gained from this qualitative study complement and expand the theory and quantitative empiricism with in-depth examinations into the experience and expectations of potential PLS users.

There are some limitations to consider when interpreting the results of this study: First, the PLSs in this study summarized psychological meta-analyses. The findings of this focus group study therefore cannot be directly applied to other disciplines due to the use of language to describe theoretical constructs and methodology specific to this field. Second, we did not recruit further focus groups until no more topics emerged (e.g., issues with certain PLS characteristics that were not mentioned before). It is thus possible that further topics would have emerged if more potential users had been interviewed. However, the aim of our study was not to develop an extensive theory, but to identify thematic aspects. Third, we did not ask actual users of the platform "KLARpsy", since this platform did not yet exist at the time of the study. The participants of this focus group study were, however, potential users who read prototypical PLSs, to help us develop the platform "KLARpsy" with its PLSs in the best possible way at this time. Fourth, this study was conducted in Germany, with PLSs in German language, and – despite efforts to optimise variation in age, sex and education level in the laypeople focus groups – the participants in the focus groups were predominantly of younger age and higher education, and in the practitioner group, only one man in comparison to five women participated. While these sample characteristics are not representative of the German population as a whole, our sample provides insight into which segments of the population such psychological PLSs may appeal to. As [35] showed, higher educated people are more likely to take part in surveys [35]. We argue that this also applies to psychological studies on plain language summaries and suggest that this might also apply to people who are likely to read psychological PLSs in general. For example, there is a possibility that it is mainly higher educated people who are interested in taking part in psychological studies on psychological plain language summaries – and that it is also these people who are more likely to read psychological PLSs. In future studies, it would be interesting to investigate how interest in such psychological PLSs can be stimulated in people who are not interested in psychological research of their own accord. How should PLSs be designed in order to arouse general interest in psychology? We also did not explicitly invite people who are currently undergoing psychotherapy or who had

undergone psychotherapy in the past to participate in our study. This is another specific user group whose needs would be interesting to explore in future studies. Nevertheless, some laypeople in our study found the PLS on psychotherapeutic interventions particularly helpful for decision-making. However, the aim of PLSs is to help people understanding scientific research rather than supporting individual decisions with a complete presentation of options and implications [8]. As this present focus group study shows, psychological practitioners considered the PLSs to be valuable informational material to be shared with interested patients. Therefore, PLSs could play an important role in the dialogue between patients and psychological practitioners, even in shared decision-making, but this still needs to be investigated in further studies [7,36].

## Conclusion

In this study, we found that the overarching aim for reading PLSs of psychological meta-analyses for all interviewed user groups is that they receive a good introduction to psychological topics.

In terms of how they expect to benefit from the PLSs, the investigated user groups differ. Interested laypeople like to gain knowledge or to use the PLS as preparation for appointments with professionals such as psychotherapists. Science communicators, on the other hand, hope to gain access to the latest scientific information on a topic, and practitioners seem to predominantly value the PLSs as useful information material for their clients or patients.

Asked for their suggestions on how the PLSs should be designed in order to be useful for them, the users had many, sometimes diverging, ideas. Therefore, to fulfill all user expectations in detail is generally difficult. Designing useful PLSs thus also means to strike a balance between meeting the range of expectations and needs of the target group.

This qualitative study complements findings of previous quantitative studies and builds upon a framework that was developed in a systematic review. It showcases how user perspectives can be integrated into the development of science communication platforms. Thus, it illustrates how evidence-based science communication that employs a participatory approach can work. With our research approach, we were able to gain deeper insights into the views of potential PLS users that go beyond our previous quantitative findings and extend the development of the platform "KLARpsy".

## Supporting information

**S1 Table. COREQ (COnsolidated criteria for REporting Qualitative research) checklist.**
(PDF)

**S2 Table. GRIPP (Guidance for Reporting Involvement of Patients and the Public) 2 Short Form Reporting checklist.**
(PDF)

**S1 File. PLSs presented in focus groups (German original & English translation).**
(PDF)

**S2 File. Interview guide.**
(PDF)

## Acknowledgments

We would like to deeply thank all participants for their time and effort invested in participating in the study. We thank Lea Stulz for her support in preparing the manuscript, and Paul Meyer-Stoll for his assistance with translating the PLSs.

## Author contributions

**Conceptualization:** Claudia Breuer, Marlene Stoll, Gesa Benz, Mark Jonas, Martin Kerwer, Anita Chasiotis.

**Data curation:** Claudia Breuer.

**Formal analysis:** Claudia Breuer, Marlene Stoll.

**Investigation:** Claudia Breuer, Marlene Stoll.

**Methodology:** Claudia Breuer, Marlene Stoll, Gesa Benz, Martin Kerwer, Anita Chasiotis.

**Project administration:** Claudia Breuer, Marlene Stoll, Anita Chasiotis.

**Software:** Claudia Breuer, Marlene Stoll.

**Writing – original draft:** Claudia Breuer, Marlene Stoll.

**Writing – review & editing:** Claudia Breuer, Marlene Stoll, Gesa Benz, Mark Jonas, Martin Kerwer, Anita Chasiotis.

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
