## [Decision Letter · Decision Letter 0]

17 Sep 2024

Dear Dr. Breuer,

Thank you for submitting your manuscript to PLOS ONE. After careful consideration, we feel that it has merit but does not fully meet PLOS ONE’s publication criteria as it currently stands. Therefore, we invite you to submit a revised version of the manuscript that addresses the points raised during the review process.

We look forward to receiving your revised manuscript.

Kind regards,

Michal Ptaszynski, PhD

Academic Editor

PLOS ONE

Journal Requirements:

3.  Thank you for stating the following financial disclosure: “This work was funded by internal funds of the Leibniz Institute for Psychology (ZPID). The authors received no third-party funding for this work.”

4. In the online submission form, you indicated that “The datasets generated and analyzed during the current study are not publicly available due to their qualitative nature that may endanger participants’ anonymity but are, in parts, available from the corresponding author on reasonable request.”

All PLOS journals now require all data underlying the findings described in their manuscript to be freely available to other researchers, either 1. In a public repository, 2. Within the manuscript itself, or 3. Uploaded as supplementary information. This policy applies to all data except where public deposition would breach compliance with the protocol approved by your research ethics board. If your data cannot be made publicly available for ethical or legal reasons (e.g., public availability would compromise patient privacy), please explain your reasons on resubmission and your exemption request will be escalated for approval.

Reviewers' comments:

Reviewer's Responses to Questions

**Comments to the Author**

1. Is the manuscript technically sound, and do the data support the conclusions?

Reviewer #1: Yes

Reviewer #2: Yes

2. Has the statistical analysis been performed appropriately and rigorously?

Reviewer #1: N/A

Reviewer #2: N/A

3. Have the authors made all data underlying the findings in their manuscript fully available?

Reviewer #1: No

Reviewer #2: No

4. Is the manuscript presented in an intelligible fashion and written in standard English?

Reviewer #1: Yes

Reviewer #2: Yes

Reviewer #1: 1. Defining the term "public" in the context of Plain Language Summaries (PLSs) would enhance the manuscript's precision and help readers understand the specific audience the authors are addressing. Therefore, I suggest that the authors define "public" immediately after the sentence: "Plain Language Summaries (PLSs) represent one approach to communicate research findings to the public."

2. The claim "One 75 core characteristic of PLSs is the use of clear and simple language to describe scientific 76 studies." needs a citation.

3. I appreciate the authors' acknowledgment that the perspective of readers on Plain Language Summaries (PLSs) in medical and health topics has been explored, while the perspective on psychological topics has not been examined in detail. However, I believe it would strengthen the manuscript if the authors provided some reasons why PLSs on medical and health topics differ from those on psychological topics. This distinction could help clarify the rationale for conducting the current study and underscore the unique aspects of psychological PLSs that warrant further investigation.

4. The manuscript states that participants received three Plain Language Summaries (PLSs) during the focus group interviews, which followed the same structure and were based on the authors' own PLS guidelines. However, it is unclear who prepared these PLSs. I suggest that the authors clarify whether the PLSs were created by the authors of the current manuscript or by the authors of the studies being summarized. Furthermore, It would be beneficial for the authors to clarify if any assessment was conducted to ensure the accuracy and reliability of the information presented in these PLSs.

5. The manuscript mentions that laypeople were recruited through announcements in German online media, but it does not specify which media outlets or platforms were used. I recommend that the authors provide more detail about the specific online media channels utilized for recruitment.

5. The inclusion criteria outlined in the manuscript raise an important question regarding the classification of participants as laypersons. While the criteria specify that laypeople should not have a degree in psychology, individuals with a master's degree in psychology, such as psychological consultants and therapists, may possess a level of familiarity with psychological jargon and concepts that could influence their perspectives on Plain Language Summaries (PLSs). I recommend that the authors provide further clarification and explanation regarding the categorization of participants. Specifically, they should address how the presence of participants with advanced degrees in psychology aligns with the study's aim to capture the layperson perspective.

6. In Table 1, which presents the characteristics of participants (N = 26), it would be beneficial for the authors to include data on the duration of each focus group.

7. The manuscript does not mention the use of a focus group protocol, which is important for enhancing the repeatability and credibility of data collection. I recommend that the authors address this omission by explaining why a formal focus group protocol was not developed or utilized in their study.

8. I am not entirely convinced that PLSs can improve the quality of research in the manner suggested in the manuscript. While the authors mention that PLSs may inspire researchers to define and sharpen their terms, I believe that the potential for PLSs to enhance research quality may be more contingent on researchers' awareness that their work will be read by a wider audience.

9. I am confused about the distinction between the categories of "Knowledge" and "Empowerment", as there appears to be some overlap in the way participants describe their experiences with PLSs. In the "Knowledge" category, participants express their desire to gain information and learn new things from the PLSs. This includes the idea that PLSs can provide valuable insights that underpin their viewpoints or enhance their understanding of specific topics. On the other hand, the "Empowerment" category focuses more on how the information gained from PLSs can facilitate decision-making, promote positive behavior changes, and support individuals in their everyday lives. This category emphasizes the practical application of knowledge, suggesting that the information from PLSs can empower individuals to take action or engage more effectively in discussions, particularly in contexts like therapy or medical appointments. To enhance clarity, I recommend that the authors provide a more explicit distinction between these two categories in the manuscript. They could elaborate on how "Knowledge" pertains to the acquisition of information, while "Empowerment" relates to the application of that knowledge in real-life situations. I guess it's not a bad idea to merge these two categories.

10. I would like to suggest a revision for the title of the category currently labeled "General Content." I believe that "Research Context" would be a more relevant and accurate title. This change would better reflect the specific concerns and insights expressed by participants regarding the contextual information necessary for understanding the research presented in the Plain Language Summaries (PLSs). The title "Research Context" would emphasize the importance of providing background information, such as details about the nationality of participants, the interventions studied, measurement methods, and other relevant factors that contribute to a comprehensive understanding of the research. This adjustment would enhance clarity and ensure that the category title accurately represents the content discussed within it. I recommend that the authors consider this change for improved precision in their presentation.

11. The authors effectively highlight the differing perspectives of laypeople, science communicators, and psychological practitioners regarding the usefulness of PLSs. It might be helpful to further elaborate on the implications of these differing perspectives for the design and dissemination of PLSs. For instance, how can the insights from each user group inform the development of PLSs that cater to their specific needs?

12. The authors do a commendable job of linking their findings to previous research. However, it would strengthen the discussion to explicitly state how the current study's findings contribute to or challenge existing literature. For example, are there any contradictions or new insights that emerge from the participants' feedback that could inform future research or practice?

13. In the methodology section, the authors should provide a rationale for choosing focus group methodology over other data collection methods, such as surveys or individual interviews.

14. The authors acknowledge that the study's findings may not be representative of the entire general population. It would be useful to expand on this point by discussing potential limitations in the sample selection and how these might affect the generalizability of the findings.

15. It is noteworthy that the focus groups did not explicitly mention the advantage of speed in reading PLSs, despite this being a significant benefit of such summaries. This omission raises questions about whether participants fully recognized or articulated this aspect during the discussions. One possible reason for this could be that participants were more focused on the content and comprehensibility of the PLSs rather than the efficiency of accessing information. Additionally, the framing of the questions or prompts used during the focus group discussions may not have encouraged participants to consider the speed of reading as a distinct advantage. It may be beneficial for the authors to reflect on this gap in the discussion section.

Reviewer #2: Congratulations to the authors on researching a very pertinent topic and doing it so well. A few comments:

- Introduction: PLSs may come in many formats, not just text. So, this can be clarified in order to educate the readers. There could be visual formats, audi or video formats, etc. And, a comment either here or in the DIscussion that those were out of scope of this research. Which in some ways could be a form of limitation.

- Methods: It would be important to understand what quality parameters were used to prepare the PLSs which were used during the focus discussions; was there a glossary of terms given that there was already awareness among the authors that the meaning of certain words could be different in psychology versus daily lives. This would give a sense of the baseline versus what the participants would expect a good PLS to be like.

- Results:

- From at least the quotes included in the main text, the Empowerment theme doesn't come through as stromgly as the rest. And, this doesn't seem to fully connect with Table 2 that lists the key themes, especially where it mentions decision aids. Was this investigated further. Are there more compelling quotes or supporting statements that indicate that indeed PLSs empower patients. One participant's comments across the categories comes across as not so positive (nothing new learnt, wouldn't read PLS again, etc). The observations noted under General content are also on similar lines that PLSs are not necessarily complete or paint a sufficiently detailed picture of research. This is also worth a comment in the Discussion that PLSs are not meant to be sources of medical advise and the risk of self-medication or stopping medication etc need careful consideration, especially given that the audience is general public. The use of decision aid even if it is for preparation for discussion with physicians is perhaps incorrect as decision aids have different quality standards, formats and recommendations, which PLSs do not necessarily need to conform to.

- As a continuation of the unmet needs highlighted by the participants, could there be another table which calls out those for developers of PLSs? There are some good nuggets of information in that cluster of quotes.

-Discussion:

- In the context of PLSs for meta-analyses or similar, it can be helpful to discuss KLARpsy in the context of efforts from Cochrane (considered to be the gold standard) which make available a library of PLSs for all their systematic reviews and meta-analysis. Are there similarities and disimilarities and learnings?

- Table 3 under Results is very informative. There are other publications which have captured similar findings with some degree of variability. It would help to discuss the findings of this study vs those.

- Were there differences in the traits between a good PLS for meta-analysis in psychology vs a good PLS for meta-analysis or other types of research in other disciplines. Can this be attributed to any inherent or perceived biases among the participants of the focus groups? To call out anything unique will be helpful for those developing psychology or meta-analysis specific PLSs.

- Are the authors considering follow-up investigations on patients or caregivers who might also be a key audience interested in findings of psychology research? If not, would that be a gap for others to investigate?

**Do you want your identity to be public for this peer review?** For information about this choice, including consent withdrawal, please see our Privacy Policy

Reviewer #1: **Yes:** Behrooz Rasuli

Reviewer #2: **Yes:** Avishek Pal and Bernice Elger

---

## [Author Response · Author response to Decision Letter 1]

12 Feb 2025

Editor Comments

E1. Please ensure that your manuscript meets PLOS ONE's style requirements, including

those for file naming. The PLOS ONE style templates can be found at

https://journals.plos.org/plosone/s/file?id=wjVg/PLOSOne_formatting_sample_main_body.pd

f and

https://journals.plos.org/plosone/s/file?id=ba62/PLOSOne_formatting_sample_title_authors_

affiliations.pdf

RE1: Thank you for this advice. We have corrected the size of the headings and

subheadings, changed the file names for the supporting information and renamed the

citation of tables in the text.

E2. We note that the grant information you provided in the ‘Funding Information’ and

‘Financial Disclosure’ sections do not match. When you resubmit, please ensure that you

provide the correct grant numbers for the awards you received for your study in the ‘Funding

Information’ section.

RE2: Thank you for this advice, but there is no grant number for this project, because it was funded internally.

E3. Thank you for stating the following financial disclosure: “This work was funded by

internal funds of the Leibniz Institute for Psychology (ZPID). The authors received no

third-party funding for this work.”Please state what role the funders took in the study. If the

funders had no role, please state: "The funders had no role in study design, data collection

and analysis, decision to publish, or preparation of the manuscript." If this statement is not

correct you must amend it as needed. Please include this amended Role of Funder

statement in your cover letter; we will change the online submission form on your behalf.

RE3: Thank you for this advice. We state that the funders had no role in study design, data

collection and analysis, decision to publish, or preparation of the manuscript.

E4. In the online submission form, you indicated that “The datasets generated and analyzed

during the current study are not publicly available due to their qualitative nature that may

endanger participants’ anonymity but are, in parts, available from the corresponding author

on reasonable request.” All PLOS journals now require all data underlying the findings

described in their manuscript to be freely available to other researchers, either 1. In a public

repository, 2. Within the manuscript itself, or 3. Uploaded as supplementary information. This

policy applies to all data except where public deposition would breach compliance with the

protocol approved by your research ethics board. If your data cannot be made publicly

available for ethical or legal reasons (e.g., public availability would compromise patient

privacy), please explain your reasons on resubmission and your exemption request will be

escalated for approval.

RE4: Thank you for pointing this out. We have added the anonymised transcripts of the

focus groups as supporting information.

Reviewer Comments

Reviewer 1:

R1.1 Defining the term "public" in the context of Plain Language Summaries (PLSs) would

enhance the manuscript's precision and help readers understand the specific audience the

authors are addressing. Therefore, I suggest that the authors define "public" immediately

after the sentence: "Plain Language Summaries (PLSs) represent one approach to

communicate research findings to the public."

RR1.1: Thank you for your suggestion. We added a definition in the introduction:

“Plain Language Summaries (PLSs) represent one approach to communicate research findings to the public.

There are also other formats using plain language, such as videos or graphics, which are sometimes called Plain

Language Resources [7]. The terminology in this field is not yet completely standardized [7; 8]. We use the term

PLS for short textual summaries of scientific studies, written in a lay-friendly manner so that lay audiences can

easily understand the described research [8; 9]. They may also benefit practitioners and researchers from other

disciplines who do not have the time or expertise knowledge needed to fully understand the scientific studies.” (l.

74-81)

R1.2. The claim "One 75 core characteristic of PLSs is the use of clear and simple language

to describe scientific 76 studies." needs a citation.

RR1.2: Thank you for this advice. We have added a citation:

“One core characteristic of PLSs is the use of clear and simple language to describe scientific studies [8].” (l. 83)

R1.3. I appreciate the authors' acknowledgment that the perspective of readers on Plain

Language Summaries (PLSs) in medical and health topics has been explored, while the

perspective on psychological topics has not been examined in detail. However, I believe it

would strengthen the manuscript if the authors provided some reasons why PLSs on medical

and health topics differ from those on psychological topics. This distinction could help clarify

the rationale for conducting the current study and underscore the unique aspects of

psychological PLSs that warrant further investigation.

RR 1.3: Thank you for this suggestion. We have added an explanation why medical and

psychological PLS differ:

“While the perspective of readers on PLSs on medical and health topics has been explored [18; 19], to our

knowledge the perspective of readers of PLSs on psychological topics has not been explored in detail. A

separate investigation of psychological PLSs is essential for at least two reasons: the distinctness of

psychological research methods; and the specificity of psychological jargon, in that the technical terms seemingly

overlap with plain language.” (l.120-125)

R1.4. The manuscript states that participants received three Plain Language Summaries

(PLSs) during the focus group interviews, which followed the same structure and were based

on the authors' own PLS guidelines. However, it is unclear who prepared these PLSs. I

suggest that the authors clarify whether the PLSs were created by the authors of the current

manuscript or by the authors of the studies being summarized. Furthermore, It would be

beneficial for the authors to clarify if any assessment was conducted to ensure the accuracy

and reliability of the information presented in these PLSs.

RR1.4: Thank you for this suggestion. We added additional information in the study design

section to make it clearer:

“During the focus group interviews, the participants received three PLSs (see S3 File). All PLSs were written by

employees of the Leibniz Institute for Psychology based on the criteria of the first version of our own PLS

guideline [28]. The guideline provided specifications for the structure and content characteristics and included

specific text modules. This ensured that all PLS were created in the same way. The specifications were

organized into five categories: linguistic characteristics, formal attributes, general content, presentation of results,

and presentation of the quality of evidence. Linguistic characteristics, for example, referred to the replacement of

technical terms with language suitable for the target audience. Formal attributes specified structural elements

such as a maximum text length. General content referred to what the PLS should include, for example, a title

related to the key message of the text. Presentation of results focused on providing in addition to quantitative

effect sizes, a qualitative explanation of the observed effects. Presentation of the quality of evidence, for

example, required a paragraph discussing potential publication bias, fostering transparency regarding the

reliability of the presented results. (l. 189-202)

R1.5a The manuscript mentions that laypeople were recruited through announcements in

German online media, but it does not specify which media outlets or platforms were used. I

recommend that the authors provide more detail about the specific online media channels

utilized for recruitment.

RR1.5a: Thank you for your recommendation. We added the different online media

channels:

“Laypeople were recruited through announcements in German online media (Facebook group for interested study

participants, eBay Kleinanzeigen, the website of the popular German science magazine “Psychologie Heute”).” (l.

206-208)

R1.5b The inclusion criteria outlined in the manuscript raise an important question regarding

the classification of participants as laypersons. While the criteria specify that laypeople

should not have a degree in psychology, individuals with a master's degree in psychology,

such as psychological consultants and therapists, may possess a level of familiarity with

psychological jargon and concepts that could influence their perspectives on Plain Language

Summaries (PLSs). I recommend that the authors provide further clarification and

explanation regarding the categorization of participants. Specifically, they should address

how the presence of participants with advanced degrees in psychology aligns with the

study's aim to capture the layperson perspective.

RR1.5b The aim of our study was to capture the perspective of various potential user groups

of PLS: laypeople, but also science communicators and psychological practitioners. PLS

may also benefit practitioners and researchers from other disciplines who do not have the

time or expertise knowledge needed to fully understand the scientific studies. We have

added the aspect in the introduction.

“They may also benefit practitioners and researchers from other disciplines who do not have the time or expertise

knowledge needed to fully understand the scientific studies.” (l. 79-81)

R1.6 In Table 1, which presents the characteristics of participants (N = 26), it would be

beneficial for the authors to include data on the duration of each focus group.

RR1.6: Unfortunately, we have only saved the durations, but not which duration corresponds

to which focus group. The recordings have already been deleted in the meantime. Only a

range and the average can be provided.

“ The focus group interviews took between 94 and 117 minutes (mean: 106 minutes).” (l.273-274)

R1.7 The manuscript does not mention the use of a focus group protocol, which is important

for enhancing the repeatability and credibility of data collection. I recommend that the

authors address this omission by explaining why a formal focus group protocol was not

developed or utilized in their study.

RR1.7: Thank you for this suggestion. The moderator used a focus group protocol which

included general instructions and a list of questions according to a semi-structured interview

guide. We clarified the process in the methods - procedure section:

“All interviews followed a focus group protocol that included general instructions for the participants and a list of

questions according to a semi-structured interview guide. The interview guide (see S4 File) was developed

beforehand and pilot-tested with non-academic staff members. The procedure was the same in all focus group

interviews.” (l.230-234)

R1.8 I am not entirely convinced that PLSs can improve the quality of research in the

manner suggested in the manuscript. While the authors mention that PLSs may inspire

researchers to define and sharpen their terms, I believe that the potential for PLSs to

enhance research quality may be more contingent on researchers' awareness that their work

will be read by a wider audience.

RR 1.8: We agree that PLS could potentially reach a broader audience and so encourage

researchers to focus more on the quality of their research. Unfortunately, this point was not

raised in the focus group. This could be due to the fact that we did not conduct a focus group

with scientists who would have most likely mentioned this point from their perspective.

R1.9 I am confused about the distinction between the categories of "Knowledge" and

"Empowerment", as there appears to be some overlap in the way participants describe their

experiences with PLSs. In the "Knowledge" category, participants express their desire to

gain information and learn new things from the PLSs. This includes the idea that PLSs can

provide valuable insights that underpin their viewpoints or enhance their understanding of

specific topics. On the other hand, the "Empowerment" category focuses more on how the

information gained from PLSs can facilitate decision-making, promote positive behavior

changes, and support individuals in their everyday lives. This category emphasizes the

practical application of knowledge, suggesting that the information from PLSs can empower

individuals to take action or engage more effectively in discussions, particularly in contexts

like therapy or medical appointments. To enhance clarity, I recommend that the authors

provide a more explicit distinction between these two categories in the manuscript. They

could elaborate on how "Knowledge" pertains to the acquisition of information, while

"Empowerment" relates to the application of that knowledge in real-life situations. I guess it's

not a bad idea to merge these two categories.

RR1.9: Thank you for these suggestions. We added the definitions of the categories in Table

2 to clarify the distinct aims. Nevertheless, we added a sentence in the result section to

make clear that the first four categories are dependent on each other:

“ The first four categories depend on each other: Accessible information is neccessary for understanding, which

can lead to an increase in knowledge and, thus, allow for informed decisions (empowerment category).” (l.

281-284)

R1.10 I would like to suggest a revision for the title of the category currently labeled

"General Content." I believe that "Research Context" would be a more relevant and accurate

title. This change would better reflect the specific concerns and insights expressed by

participants regarding the contextual information necessary for understanding the research

presented in the Plain Language Summaries (PLSs). The title "Research Context" would

emphasize the importance of providing background information, such as details about the

nationality of participants, the interventions studied, measurement methods, and other

relevant factors that contribute to a comprehensive understanding of the research. This

adjustment would enhance clarity and ensure that the category title accurately represents

the content discussed within it. I recommend that the authors consider this change for

improved precision in their presentation.

RR. 1.10: Thanks for this suggestion. Indeed, most of the aspects mentioned in this category

refer to research, but there are also some other aspects that are more related to the

alignment of content. Therefore, we believe the term “Research context” would not cover

these aspects. We have added the definition of “General Content” from Stoll et al. in Table 2

to clarify this.

R1.11 The authors effectively highlight the differing perspectives of laypeople, science

communicators, and psychological practitioners regarding the usefulness of PLSs. It might

be helpful to further elaborate on the implications of these differing perspectives for the

design and dissemination of PLSs. For instance, how can the insights from each user group

inform the development of PLSs that cater to their specific needs?

RR1.11: Thank you for pointing this out. We have divided the characteristics for PLSs in

table 3 according to the different user groups and have adjusted the text in Section 3.2

accordingly.

R1.12 The authors do a commendable job of linking their findings to previous research.

However, it would strengthen the discussion to explicitly state how the current study's

findings contribute to or challenge existing literature. For example, are there any

contradictions or new insights that emerge from the participants' feedback that could inform

future research or practice?

RR1.12: Thank you for this advice. We added a paragraph on the comparison with existing

literature in the main findings section.

“While the subject of this study was PLSs on psychological studies, the subject of the majority of previous

research on the user experience of PLSs has been PLSs on medical studies. Our study indicates that users’

needs in terms of the design of the PLSs are similar between psychological and medical PLSs as determined in

previous research. The extent to which there are specific differences in t

---

## [Decision Letter · Decision Letter 1]

18 Jun 2025

Dear Dr. Breuer,

Thank you for submitting your manuscript to PLOS ONE. After careful consideration, we feel that it has merit but does not fully meet PLOS ONE’s publication criteria as it currently stands. Therefore, we invite you to submit a revised version of the manuscript that addresses the points raised during the review process.

We look forward to receiving your revised manuscript.

Kind regards,

Muhammad Zammad Aslam, Ph.D.

Academic Editor

PLOS ONE

Journal Requirements:

Reviewers' comments:

Reviewer's Responses to Questions

**Comments to the Author**

Reviewer #3: (No Response)

Reviewer #4: All comments have been addressed

2. Is the manuscript technically sound, and do the data support the conclusions?

Reviewer #3: Partly

Reviewer #4: Partly

3. Has the statistical analysis been performed appropriately and rigorously?

Reviewer #3: N/A

Reviewer #4: Yes

4. Have the authors made all data underlying the findings in their manuscript fully available?

Reviewer #3: No

Reviewer #4: Yes

5. Is the manuscript presented in an intelligible fashion and written in standard English?

Reviewer #3: No

Reviewer #4: Yes

Reviewer #3: Thank you for the opportunity to review this revised version of your manuscript. I can see from the previous reviewer comments, you have put a great deal of time and effort into addressing reviewer comments and concerns. Despite this, I do not think the manuscript is at a point where it is able to be published in its current form. Below I have outlined my areas of concern. Please note that although my feedback might be perceived as negative, that does not underscore the importance I place on this study and the body of work in this field, both that conducted by the authors and by others.

Abstract

• Mentions general population, then laypeople. Inconsistent terminology.

• “We wanted to know what the aims and benefits of reading psychological PLSs for them are and how these PLSs should be designed, in their point of view, in order to be useful.” – poor grammar.

• Mixed tense.

• This abstract does not give me a good understanding of the manuscript, nor is it the best representation of the study. I would recommend re-writing it.

Introduction

• Need reference for study mentioned lines 57-59. Assume it is reference 2, but need to include for both sentences about this study.

• I don’t understand the use of reference number 5 and the example of psychology and climate change in the context of communicating to the public – after reading this paper (reference 5), it does not seem to hold appeal for a broad audience.

• Line 74 – need to clarify that you are discussing text-based PLSs in the opening sentence.

• Line 82 – Autism is not a discipline – it is a topic within the field of health and medicine.

• Line 84-86 – This sentence needs a reference “This is especially challenging for psychological research as the jargon of psychological researchers often sounds like plain language, but is actually a technical term (e.g., “motivation”).” Motivation should not be in quotation marks within the brackets.

• Line 95 – this is poor grammar – “The leading question for the project was “How should a good PLS of a psychological meta-analysis look like?”

• The authors need to keep referencing their previous work throughout the paragraph, after referencing it on line 104.

• Referring to comment 1.3 by reviewer 1, more details is need on what it is that makes psychological research methods distinct?

Methods

• Line 184 - Why are the authors using reference 27 and mentioning the use of focus groups in the context of market research, when the context for this study is scientific research.

• Did someone from the research team check the PLSs to ensure they were written according to the guidelines provided before they were used in the interviews?

• It would be useful to understand the participant selection process more fully to ensure there was no bias i.e., how were participants selected from the pool of respondents i.e., how they aimed “to maximize variation of socio-demographic characteristics (age, sex, education level) in the sample”. Were group numbers for each cohort determined prior to recruitment?

• Line 192 – I would have liked to review the three PLSs that were part of the interviews, however S3 file was only provided in German. I am fluent in English only.

• Line 245 - Was there a reason these topics were chosen for the three PLSs? Given that it is stated that each focus group session ended with participants being asked about their willingness to recommend “this type of PLS”, I am wondering if the PLSs were chosen for a specific reason. Also, what do the authors mean by “type” of PLS? Format, length, topic etc.

• Line 255 - It would be useful for the authors to include their rationale for their method of data analysis based on reference 29. Also, a short description of this method would be helpful as many readers might not be familiar with it.

Results

• Line 273 – I would like to see a summary of the key results, not details about the focus groups, some of which belongs in the Methods section. Authors being with 3.1. User’s aims without any introduction.

• Table 1 – Participant characteristics. I suggest the authors remember that this paper is likely to appeal to an international audience, so providing context for education levels would be helpful.

• My concern is that interviews were constrained by the framework the authors developed and reported in a previous systematic review (reference 8).

• Line 502 – what are the authors reporting on aspects of the PLSs that the participants “missed”? This does not appear to be in alignment with the study aims as this is an evaluation of the ability of the participants to understand the information in the PLS. No criteria for this type of rating was established in the Methods.

Discussion

• Line 637 – how is psychological research different to medical research? Isn’t psychological research a sub-set of medical research?

• Line 644 – this should be under the sub-heading of Strengths and Weaknesses

• Line 678 – I do not think the authors can state that the study sample was diverse, nor that diversity of the sample was a strength of the study, since participants were all recruited from Germany. Authors took steps to create diversity, however, their claim of diversity is overstated. Beginning line 701, authors outline the limited diversity of the sample as a study limitation, so there is clear confusion about this issue.

• Lines 684 – 687 – the first sentence is non-sensical and the following sentence is not a particular strength as it is standard practice for qualitative research.

• Line 692 – authors mention saturation for the first time. This should have been addressed in the Methods.

• Line 697 – If KLARpsy did not exist at the time of the study, why is it mentioned in line 97 of the Introduction? Authors should correct this inconsistency.

• Line 707 – This statement is unsupported - “For example, it is possible that people with a lower level of education did not feel motivated to participate in the study”.

Reviewer #4: The authors have addressed the raised questions and revised the manuscript well. The manuscript presents a well-structured and comprehensive focus group study exploring users' perspectives on German Plain Language Summaries (PLS) of psychological meta-analyses.

**Do you want your identity to be public for this peer review?** For information about this choice, including consent withdrawal, please see our Privacy Policy

Reviewer #3: No

Reviewer #4: **Yes:** Ushba Rasool

---

## [Author Response · Author response to Decision Letter 2]

15 Aug 2025

All comments of the reviewers were answered point-by point below. Given page numbers and sections, refer to the version with track changes.

• Mentions general population, then laypeople. Inconsistent terminology.

In the revised version of the abstract, we have replaced the term “general population” with “lay audiences” (p.2, l.28).

• Abstract - “We wanted to know what the aims and benefits of reading psychological PLSs for them are and how these PLSs should be designed, in their point of view, in order to be useful.” – poor grammar.

We have changed the sentence into: “We wanted to understand the aims and benefits of reading psychological PLSs from their perspective, and how these PLSs should be designed to be useful to them.” (p.2, l.31-33)

• Abstract - Mixed tense

In the revised version of the abstract, we now avoid mixed tenses.

• Abstract - This abstract does not give me a good understanding of the manuscript, nor is it the best representation of the study. I would recommend re-writing it

We appreciate the reviewer’s comment. We have rewritten the abstract to provide a better understanding of the manuscript.

• Need reference for study mentioned lines 57-59. Assume it is reference 2, but need to include for both sentences about this study.

We cite reference 2 in both sentences now (p.3, l.67)

• I don’t understand the use of reference number 5 and the example of psychology and climate change in the context of communicating to the public – after reading this paper (reference 5), it does not seem to hold appeal for a broad audience

The reference number 5 is cited for the statement that “psychological evidence becomes increasingly important (...) to deal with problems like climate change that affect society as a whole.” Here, we do not want to make the point that the findings currently are communicated to the public (which they are often not) but that they provide solutions for problems that affect all of us and therefore should be shared with society as a whole in the future, and not only with scientists. We rephrased the sentence so that our point becomes clearer (p.3, l.71-74).

• Line 74 – need to clarify that you are discussing text-based PLSs in the opening sentence.

We have changed the sentence to point out that PLSs are text-based: “Text-based Plain Language Summaries (PLSs) represent one approach for communicating research findings to the public.” (p.4, l.84-85)

• Line 82 – Autism is not a discipline – it is a topic within the field of health and medicine

Thank you for this advice. We have replaced “disciplines“ with „fields and topics“ (p.4, l. 91-92)

• Line 84-86 – This sentence needs a reference “This is especially challenging for psychological research as the jargon of psychological researchers often sounds like plain language, but is actually a technical term (e.g., “motivation”).” Motivation should not be in quotation marks within the brackets.

We added a reference for this sentence (p.4, l. 97).

• Line 95 – this is poor grammar – “The leading question for the project was “How should a good PLS of a psychological meta-analysis look like?”

Thank you for pointing this out. We have improved the grammar of this sentence (p. 5, l. 106).

• The authors need to keep referencing their previous work throughout the paragraph, after referencing it on line 104.

Reference 8 is cited in all relevant sentences now (p.5, l. 116, 120, 122).

• Referring to comment 1.3 by reviewer 1, more details is need on what it is that makes psychological research methods distinct?

We have added a more detailed explanation: “A separate investigation of psychological PLSs is essential for at least two reasons: first, the distinctness of psychological research: as the subject is to examine the mind, the constructs of interest are often not directly observable, but must be inferred or operationalized by manipulating proxy variables. Furthermore, studies often rely on correlation designs and context-sensitive interpretations. Thus, psychology is a very nuanced and complex field with a growing complexity of research methods. Since it is a relatively new field, methods and subdisciplines are evolving rapidly. And second, the specificity of psychological jargon, in that the technical terms seemingly overlap with plain language.” (p.6, l. 134-142)

• Line 184 - Why are the authors using reference 27 and mentioning the use of focus groups in the context of market research, when the context for this study is scientific research

Thank you for pointing this out. The respective sentence has been removed.

• Did someone from the research team check the PLSs to ensure they were written according to the guidelines provided before they were used in the interviews?

As is written in the text, the “PLSs were written (...) based on the criteria of the first version of our own PLS guideline”. They were written by the study authors themselves, which was not mentioned in the text. We added this information to clarify in the text: “All PLSs were written in German based on the criteria of the first version of the PLS guideline [28] by AC, MK, MS and GB, who are employees of the Leibniz Institute for Psychology and who have experience with writing these kind of PLSs.” (p.8, l.208-11)

• It would be useful to understand the participant selection process more fully to ensure there was no bias i.e., how were participants selected from the pool of respondents i.e., how they aimed “to maximize variation of socio-demographic characteristics (age, sex, education level) in the sample”. Were group numbers for each cohort determined prior to recruitment?

We describe the recruitment and selection process more clearly now:

1. We have stated that the number of groups was determined before recruitment started: “We aimed to conduct two focus group interviews with laypeople, one focus group interview with science communicators, and one with psychologists working as counselor and/or psychotherapists (“psychological practitioners”) (p.8, l. 202-04)

2. We have provided more details on the recruitment process in S5 table.

3. We have explained why we had to select participants who fulfilled the inclusion criteria: “For laypeople, expressions of interest exceeded the number of available spots, so one researcher (CB) selected participants with the aim of maximizing variation in socio-demographic characteristics (age, sex, education level) within this group”. (p.10, l.240-43)

• Line 192 – I would have liked to review the three PLSs that were part of the interviews, however S3 file was only provided in German. I am fluent in English only.

We have now provided an English translation of the three PLS in S4 File. Please note that this is a translated version of the original PLSs provided for informational purposes only. Due to the nature of plain language summaries, the exact content and complexity (compared to scientific language) of the original cannot be fully reproduced in translation.

• Line 245 - Was there a reason these topics were chosen for the three PLSs? Given that it is stated that each focus group session ended with participants being asked about their willingness to recommend “this type of PLS”, I am wondering if the PLSs were chosen for a specific reason. Also, what do the authors mean by “type” of PLS? Format, length, topic etc.

Thank you for pointing this out. Prior to this question at the end, the moderator had explained that the group discussion was about PLSs of psychological meta-analyses and that the PLSs presented were prototypes. Moreover, a website with many PLSs on different psychological topics will be developed subsequently. The question „would you recommend this kind of PLS?“ referred to those psychological PLSs in the form which they had been presented. The topics were chosen so that they cover different areas of psychological research (clinical research, general psychology, mental health / individual differences).

• Line 255 - It would be useful for the authors to include their rationale for their method of data analysis based on reference 29. Also, a short description of this method would be helpful as many readers might not be familiar with it.

We now provide a rationale for choosing the approach according to Kuckartz and Rädiker and have explained the method using our analysis example (p.13, l. 288-300)

• Line 273 – I would like to see a summary of the key results, not details about the focus groups, some of which belongs in the Methods section. Authors being with 3.1. User’s aims without any introduction.

Thank you for this suggestion. We have integrated the introductory paragraph into section 2.2 Setting and participants (p.10, l.245-49).

• Table 1 – Participant characteristics. I suggest the authors remember that this paper is likely to appeal to an international audience, so providing context for education levels would be helpful.

To provide context without going beyond the scope, we have classified educational levels now as low, moderate and high, and summarised the categories qualification for university and university degree in Table 1.

• My concern is that interviews were constrained by the framework the authors developed and reported in a previous systematic review (reference 8).

Thank you for pointing this out. For our analysis, we chose a deductive-inductive categorization process according to Kuckartz and Rädiker. This means we based the analysis on a previously developed categorisation system, but allowed additional categories and subcategories to be derived from our data. In other words, we used the framework from Stoll et al. only as a starting point, but inductive building of main and subcategories was possible and was done for subcategories. Moreover, our interview guide contained only open questions on the aims and characteristics of PLSs.

• Line 502 – what are the authors reporting on aspects of the PLSs that the participants “missed”? This does not appear to be in alignment with the study aims as this is an evaluation of the ability of the participants to understand the information in the PLS. No criteria for this type of rating was established in the Methods.

Thank you for your comment. Our second research question was „How should PLSs, from the users’s point of view, be designed in order to be useful?“ We explained at the beginning of each discussion that the PLSs presented during the discussion are text-based summaries of psychological research. During the interviews we explicitly asked participants which characteristics they find important for psychological PLSs from their point of view. We only analysed participants‘ comments and did not assess their ability to understand the information. We have included a participant’s quote to support the statement (p.27, l.570-71).

When revisiting this paragraph we realized that there was some ambiguity here and that our sentence might have been misleading. We, thus, clarified the sentence so that it becomes clearer what we mean: “Participants of all groups pointed out that a general statement or ranking about reliability of results, which would help them to assess the meaningfulness and trustworthiness of the results, was lacking.” (p.7, l. 567- 69)

• Line 637 – how is psychological research different to medical research? Isn’t psychological research a sub-set of medical research?

Psychological research is not a sub-discipline of medical research but a discipline in its own right. The subject is human experience, cognitions and behaviour. There are some overlaps between psychiatric medical research and research in clinical psychology when pathological behaviour is investigated - but clinical psychology is only one out of many sub-disciplines of psychology.

• Line 644 – this should be under the sub-heading of Strengths and Weaknesses

Thank you for pointing this out. The aspect of representativeness is already mentioned under 4.3 Strengths and Limitations, so we have removed it from 4.2 Main findings in Context.

• Line 678 – I do not think the authors can state that the study sample was diverse, nor that diversity of the sample was a strength of the study, since participants were all recruited from Germany. Authors took steps to create diversity, however, their claim of diversity is overstated. Beginning line 701, authors outline the limited diversity of the sample as a study limitation, so there is clear confusion about this issue.

Thank you for your comment. We removed the sentence „A strength of this study is the diversity of our sample.“ from section 4.3 Strengths and Limitations to avoid confusion. However, it is important for us to emphasize that in this case it does not make sense to relate diversity to the place of residence of the participants. The target group of the texts are persons who can speak German as the texts are in German. Interestingly, one participant explicitly appreciated the fact that the information was provided in their native language rather than in English during the interviews.

• Lines 684 – 687 – the first sentence is non-sensical and the following sentence is not a particular strength as it is standard practice for qualitative research.

We removed both sentences from section 4.3

• Line 692 – authors mention saturation for the first time. This should have been addressed in the Methods.

We have addressed saturation in the methods (p.8, l. 206).

• Line 697 – If KLARpsy did not exist at the time of the study, why is it mentioned in line 97 of the Introduction? Authors should correct this inconsistency.

Thank you for pointing this out. We have removed the sentence in line 107 to avoid confusion. However, the development of KLARpsy is still mentioned in section 4.2 Main findings in context (p. 35, l. 698).

• Line 707 – This statement is unsupported - “For example, it is possible that people with a lower level of education did not feel motivated to participate in the study”.

We reformulated this sentence so that it is clear that this is only one possible hypothetical explanation: “For example, there is a possibility that it is mainly higher educated people who are interested in taking part in psychological studies on psychological plain language summaries – and that it is also these people who are more likely to read psychological PLSs.” (p.37, l.758)

---

## [Decision Letter · Decision Letter 2]

11 Sep 2025

Dear Dr. Breuer,

We look forward to receiving your revised manuscript.

Kind regards,

Muhammad Zammad Aslam, Ph.D.

Academic Editor

PLOS ONE

Journal Requirements:

**Additional Editor Comments:**

Dear authors, please see Reviewer 3's comments, which suggest minor revisions. Regards

Reviewers' comments:

Reviewer's Responses to Questions

**Comments to the Author**

Reviewer #3: (No Response)

2. Is the manuscript technically sound, and do the data support the conclusions?

Reviewer #3: Yes

3. Has the statistical analysis been performed appropriately and rigorously?

Reviewer #3: Yes

4. Have the authors made all data underlying the findings in their manuscript fully available?

Reviewer #3: Yes

5. Is the manuscript presented in an intelligible fashion and written in standard English?

Reviewer #3: Yes

Reviewer #3: Reviewer Comments to revision

Thank you for the opportunity to review this revised manuscript. I believe it still required some work before it is ready for publication. For those instances where the authors have not fully addressed my previous comments, I have included both below, along with my follow up comment

Reviewer comments and responses to the original manuscript

1. Reviewer comment - original

I don’t understand the use of reference number 5 and the example of psychology and climate change in the context of communicating to the public – after reading this paper (reference 5), it does not seem to hold appeal for a broad audience

Author response

The reference number 5 is cited for the statement that “psychological evidence becomes increasingly important (...) to deal with problems like climate change that affect society as a whole.” Here, we do not want to make the point that the findings currently are communicated to the public (which they are often not) but that they provide solutions for problems that affect all of us and therefore should be shared with society as a whole in the future, and not only with scientists. We rephrased the sentence so that our point becomes clearer (p.3, l.71-74).

Reviewer comment – follow up

I appreciate the authors’ response, however, it does not address my comment, and the sentence is still not clear. Perhaps the authors could expand on how psychological findings help society deal with problems such as climate change.

2. Reviewer comment

Referring to comment 1.3 by reviewer 1, more details is need on what it is that makes psychological research methods distinct?

Author response

We have added a more detailed explanation: “A separate investigation of psychological PLSs is essential for at least two reasons: first, the distinctness of psychological research: as the subject is to examine the mind, the constructs of interest are often not directly observable, but must be inferred or operationalized by manipulating proxy variables. Furthermore, studies often rely on correlation designs and context-sensitive interpretations. Thus, psychology is a very nuanced and complex field with a growing complexity of research methods. Since it is a relatively new field, methods and subdisciplines are evolving rapidly. And second, the specificity of psychological jargon, in that the technical terms seemingly overlap with plain language.” (p.6, l. 134-142)

Reviewer comment – follow up

I appreciate this detailed response. A minor note – authors state that there are two reasons, however I have located four separate points, all valid in their own right. For example:

1. the distinctness of psychological research: as the subject is to examine the mind, the constructs of interest are often not directly observable, but must be inferred or operationalized by manipulating proxy variables.

2. Furthermore, studies often rely on correlation designs and context-sensitive interpretations

3. psychology is a very nuanced and complex field with a growing complexity of research methods. Since it is a relatively new field, methods and subdisciplines are evolving rapidly.

4. the specificity of psychological jargon, in that the technical terms seemingly overlap with plain language

Re-writing this paragraph for clarity would be helpful. It should also be referenced throughout.

3. Reviewer comment - original

It would be useful to understand the participant selection process more fully to ensure there was no bias i.e., how were participants selected from the pool of respondents i.e., how they aimed “to maximize variation of socio-demographic characteristics (age, sex, education level) in the sample”. Were group numbers for each cohort determined prior to recruitment?

Author response

We describe the recruitment and selection process more clearly now:

1. We have stated that the number of groups was determined before recruitment started: “We aimed to conduct two focus group interviews with laypeople, one focus group interview with science communicators, and one with psychologists working as counselor and/or psychotherapists (“psychological practitioners”) (p.8, l. 202-04)

2. We have provided more details on the recruitment process in S5 table.

3. We have explained why we had to select participants who fulfilled the inclusion criteria: “For laypeople, expressions of interest exceeded the number of available spots, so one researcher (CB) selected participants with the aim of maximizing variation in socio-demographic characteristics (age, sex, education level) within this group”. (p.10, l.240-43)

Reviewer comment – follow up

1. Addressed with thanks.

2. Thank for the additional detail regarding recruitment. I note on p9 authors state “The detailed recruitment strategy is reported in S5 Table.” I find the level of information in this table is lacking in detail. I do not understand why this information was removed from the Methods section and put in a supplementary table. It could easily be summarised in a few sentences in the Methods section.

3. Please refer to my original question, in which I asked ‘how’, not ‘why’ were participants selected from the pool or respondents i.e., i.e., how did authors aim “to maximize variation of socio-demographic characteristics (age, sex, education level) in the sample”

4. Reviewer comment - original

Table 1 – Participant characteristics. I suggest the authors remember that this paper is likely to appeal to an international audience, so providing context for education levels would be helpful.

Author response

To provide context without going beyond the scope, we have classified educational levels now as low, moderate and high, and summarised the categories qualification for university and university degree in Table 1.

Reviewer comment – follow up

I appreciate the effort the reclassify educational qualifications, however, I do not think that the using low, moderate and high is a good choice in terms of language. It could be alienating for some readers.

I also note that Table 1 appears to have moved to the Methods section. Since it reports study findings, it should be in the Results section.

5. Reviewer comment - original

Line 192 – I would have liked to review the three PLSs that were part of the interviews, however S3 file was only provided in German. I am fluent in English only.

Author response

We have now provided an English translation of the three PLS in S4 File. Please note that this is a translated version of the original PLSs provided for informational purposes only. Due to the nature of plain language summaries, the exact content and complexity (compared to scientific language) of the original cannot be fully reproduced in translation.

Reviewer comment – follow up

Thank you for providing the PLSs translated into English as requested. I would have also liked to see the guidelines in English, but acknowledge that I did not request these.

6. Reviewer comment - original

Line 707 – This statement is unsupported - “For example, it is possible that people with a lower level of education did not feel motivated to participate in the study”.

Author response

We reformulated this sentence so that it is clear that this is only one possible hypothetical explanation: “For example, there is a possibility that it is mainly higher educated people who are interested in taking part in psychological studies on psychological plain language summaries – and that it is also these people who are more likely to read psychological PLSs.” (p.37, l.758)

Reviewer comment – follow up

I appreciate that this statement has been reworded, however it still needed to be cited with supporting evidence.

General comment

• When introducing the acronym (PLSs) for plain language summaries, all letters should be lower case, unless used at the start of a sentence i.e., plain language summaries (PLSs). This is in keeping with the established convention of other published research in the field, including the cited work by Stoll et al., 2022.

**Do you want your identity to be public for this peer review?** For information about this choice, including consent withdrawal, please see our Privacy Policy

Reviewer #3: **Yes:** Karen Gainey

---

## [Author Response · Author response to Decision Letter 3]

2 Dec 2025

1. Reviewer comment - original

I don’t understand the use of reference number 5 and the example of psychology and climate change in the context of communicating to the public – after reading this paper (reference 5), it does not seem to hold appeal for a broad audience

Author response

The reference number 5 is cited for the statement that “psychological evidence becomes increasingly important (...) to deal with problems like climate change that affect society as a whole.” Here, we do not want to make the point that the findings currently are communicated to the public (which they are often not) but that they provide solutions for problems that affect all of us and therefore should be shared with society as a whole in the future, and not only with scientists. We rephrased the sentence so that our point becomes clearer (p.3, l.71-74).

Reviewer comment – follow up

I appreciate the authors’ response, however, it does not address my comment, and the sentence is still not clear. Perhaps the authors could expand on how psychological findings help society deal with problems such as climate change.

1. Authors' response: We reviewed this paragraph and reformulated it so that it hopefully becomes clearer. Also, we added another reference of the American Psychological Association on the interface between psychology and global climate change (p. 3, l. 63-70).

2.1 2. Reviewer comment

Referring to comment 1.3 by reviewer 1, more details is need on what it is that makes psychological research methods distinct?

Author response

We have added a more detailed explanation: “A separate investigation of psychological PLSs is essential for at least two reasons: first, the distinctness of psychological research: as the subject is to examine the mind, the constructs of interest are often not directly observable, but must be inferred or operationalized by manipulating proxy variables. Furthermore, studies often rely on correlation designs and context-sensitive interpretations. Thus, psychology is a very nuanced and complex field with a growing complexity of research methods. Since it is a relatively new field, methods and subdisciplines are evolving rapidly. And second, the specificity of psychological jargon, in that the technical terms seemingly overlap with plain language.” (p.6, l. 134-142)

Reviewer comment – follow up

I appreciate this detailed response. A minor note – authors state that there are two reasons, however I have located four separate points, all valid in their own right. For example:

1. the distinctness of psychological research: as the subject is to examine the mind, the constructs of interest are often not directly observable, but must be inferred or operationalized by manipulating proxy variables.

2. Furthermore, studies often rely on correlation designs and context-sensitive interpretations

3. psychology is a very nuanced and complex field with a growing complexity of research methods. Since it is a relatively new field, methods and subdisciplines are evolving rapidly.

4. the specificity of psychological jargon, in that the technical terms seemingly overlap with plain language

Re-writing this paragraph for clarity would be helpful. It should also be referenced throughout.

2. Authors' response: Thank you for your feedback. We re-wrote the paragraph to improve clarity and added references (p. 5, l. 127-142).

3. Reviewer comment - original

It would be useful to understand the participant selection process more fully to ensure there was no bias i.e., how were participants selected from the pool of respondents i.e., how they aimed “to maximize variation of socio-demographic characteristics (age, sex, education level) in the sample”. Were group numbers for each cohort determined prior to recruitment?

Author response

We describe the recruitment and selection process more clearly now:

1. We have stated that the number of groups was determined before recruitment started: “We aimed to conduct two focus group interviews with laypeople, one focus group interview with science communicators, and one with psychologists working as counselor and/or psychotherapists (“psychological practitioners”) (p.8, l. 202-04)

2. We have provided more details on the recruitment process in S5 table.

3. We have explained why we had to select participants who fulfilled the inclusion criteria: “For laypeople, expressions of interest exceeded the number of available spots, so one researcher (CB) selected participants with the aim of maximizing variation in socio-demographic characteristics (age, sex, education level) within this group”. (p.10, l.240-43)

Reviewer comment – follow up

1. Addressed with thanks.

2. Thank for the additional detail regarding recruitment. I note on p9 authors state “The detailed recruitment strategy is reported in S5 Table.” I find the level of information in this table is lacking in detail. I do not understand why this information was removed from the Methods section and put in a supplementary table. It could easily be summarised in a few sentences in the Methods section.

3. Please refer to my original question, in which I asked ‘how’, not ‘why’ were participants selected from the pool or respondents i.e., i.e., how did authors aim “to maximize variation of socio-demographic characteristics (age, sex, education level) in the sample”

3. Authors' response:

Thank you for your comment. We included the information of S5 table in the text again (p. 9, l. 224-233).

Regarding the selection process, we describe the process more clearly now: “After voluntarily expressing their interest to participate via e-mail, potential participants received an informational letter on the procedure and conditions, and a consent form along with a short questionnaire on inclusion criteria and sociodemographic characteristics (age, gender, education level). Inclusion criteria were sufficient German language skills, legal age, interest in psychology and no degree in psychology (only laypeople) or a master’s degree in psychology and being psychological counselors and/or psychological therapists (only psychological practitioners) or professional involvement with science communication including psychological topics (only science communicators). No other characteristics were collected that could indicate any attitudes toward PLSs. For laypeople, the number of submitted consent forms exceeded the number of available spots. While we did not employ stratified sampling or use a recruitment panel, we aimed to maximize variation in socio-demographic characteristics (i.e. a balanced gender distribution, different age decades and inclusion of the few participants with lower education levels) by purposively selecting individuals to achieve a heterogeneous sample. For psychological practitioners or science communicators, the number of submitted consent forms did not exceed the number of available spots.” (p. 9, l. 235-251)

4. Reviewer comment - original

Table 1 – Participant characteristics. I suggest the authors remember that this paper is likely to appeal to an international audience, so providing context for education levels would be helpful.

Author response

To provide context without going beyond the scope, we have classified educational levels now as low, moderate and high, and summarised the categories qualification for university and university degree in Table 1.

Reviewer comment – follow up

I appreciate the effort the reclassify educational qualifications, however, I do not think that the using low, moderate and high is a good choice in terms of language. It could be alienating for some readers.

I also note that Table 1 appears to have moved to the Methods section. Since it reports study findings, it should be in the Results section.

4. Authors' response:

Thank you for your comment. We partially reclassified the education levels according to the UNESCO International Standard Classification of Education. We have also rearranged Table 1 in the Results section and included a new short paragraph titled “Participants’ characteristics” (p. 13, l. 310-14).

5. Reviewer comment - original

Line 192 – I would have liked to review the three PLSs that were part of the interviews, however S3 file was only provided in German. I am fluent in English only.

Author response

We have now provided an English translation of the three PLS in S4 File. Please note that this is a translated version of the original PLSs provided for informational purposes only. Due to the nature of plain language summaries, the exact content and complexity (compared to scientific language) of the original cannot be fully reproduced in translation.

Reviewer comment – follow up

Thank you for providing the PLSs translated into English as requested. I would have also liked to see the guidelines in English, but acknowledge that I did not request these.

5. Authors' response:

We appreciate the reviewer’s interest in the guideline. We fully agree that making it available in English would be valuable. However, due to time constraints, we are unable to provide an English version at this stage. We plan to translate the guideline and make it publicly available next year.

6. Reviewer comment - original

Line 707 – This statement is unsupported - “For example, it is possible that people with a lower level of education did not feel motivated to participate in the study”.

Author response

We reformulated this sentence so that it is clear that this is only one possible hypothetical explanation: “For example, there is a possibility that it is mainly higher educated people who are interested in taking part in psychological studies on psychological plain language summaries – and that it is also these people who are more likely to read psychological PLSs.” (p.37, l.758)

Reviewer comment – follow up

I appreciate that this statement has been reworded, however it still needed to be cited with supporting evidence.

6. Authors' response: We added a citation of a study investigating education bias to strengthen our reasoning (p. 36, l. 734-737).

7. When introducing the acronym (PLSs) for plain language summaries, all letters should be lower case, unless used at the start of a sentence i.e., plain language summaries (PLSs). This is in keeping with the established convention of other published research in the field, including the cited work by Stoll et al., 2022.

7. Authors' response:

The term is written now in lowercase when we introduce it (p. 4, l. 80).

---

## [Decision Letter · Decision Letter 3]

17 Dec 2025

Dear Dr. Breuer,

Thank you for submitting your manuscript to PLOS ONE. After careful consideration, we feel that it has merit but does not fully meet PLOS ONE’s publication criteria as it currently stands. Therefore, we invite you to submit a revised version of the manuscript that addresses the points raised during the review process.

We look forward to receiving your revised manuscript.

Kind regards,

Muhammad Zammad Aslam, Ph.D.

Academic Editor

PLOS One

Journal Requirements:

Additional Editor Comments:

Dear Authors,

Thank you for submitting the revised version of your manuscript. We appreciate the time and effort invested in addressing the reviewers’ comments.

Following further evaluation, the reviewer has indicated that several important concerns remain unresolved. In particular, there are instances where statements are presented without appropriate supporting references, and some sections would benefit from improved clarity.

The reviewer has also noted that certain comments raised in previous rounds do not appear to have been fully addressed. In some cases, the response indicates that changes were made; however, these revisions are not clearly identifiable within the manuscript. To facilitate further assessment, it is essential that all revisions are explicitly and transparently documented.

Should you wish to submit a further revision, please ensure the following:

1. Provide a detailed, point-by-point response to each reviewer comment, clearly indicating how and where each issue has been addressed.

2. Use tracked changes (or clearly highlighted text) in the manuscript so that all revisions are easily identifiable.

3. Ensure that all statements requiring support are appropriately referenced, and that references are current and relevant.

4. Revise for clarity, particularly in sections previously identified as unclear.

5. Where a reviewer comment has not been addressed, provide a clear and reasoned explanation in the response document.

Careful attention to these matters is essential to meet publication standards and uphold research integrity. A thorough and transparent revision will be necessary for further consideration of the manuscript.

Thank you for your understanding and cooperation. We look forward to receiving a carefully revised submission should you decide to proceed.

Kind regards

Reviewers' comments:

Reviewer's Responses to Questions

**Comments to the Author**

Reviewer #3: (No Response)

2. Is the manuscript technically sound, and do the data support the conclusions?

Reviewer #3: Yes

3. Has the statistical analysis been performed appropriately and rigorously?

Reviewer #3: Yes

4. Have the authors made all data underlying the findings in their manuscript fully available?

Reviewer #3: Yes

5. Is the manuscript presented in an intelligible fashion and written in standard English?

Reviewer #3: No

Reviewer #3: I have provided my deteiled responses to all items in an attached file labelled "Reviewer Comments to revision".

**Do you want your identity to be public for this peer review?** For information about this choice, including consent withdrawal, please see our Privacy Policy

Reviewer #3: **Yes:** Dr Karen Gainey

---

## [Author Response · Author response to Decision Letter 4]

2 Feb 2026

Item 1

Reviewer comment – revision 1

I don’t understand the use of reference number 5 and the example of psychology and climate change in the context of communicating to the public – after reading this paper (reference 5), it does not seem to hold appeal for a broad audience

Author response

The reference number 5 is cited for the statement that “psychological evidence becomes increasingly important (...) to deal with problems like climate change that affect society as a whole.” Here, we do not want to make the point that the findings currently are communicated to the public (which they are often not) but that they provide solutions for problems that affect all of us and therefore should be shared with society as a whole in the future, and not only with scientists. We rephrased the sentence so that our point becomes clearer (p.3, l.71-74).

Reviewer comment – revision 2

I appreciate the authors’ response, however, it does not address my comment, and the sentence is still not clear. Perhaps the authors could expand on how psychological findings help society deal with problems such as climate change.

Authors' response

We reviewed this paragraph and reformulated it so that it hopefully becomes clearer. Also, we added another reference of the American Psychological Association on the interface between psychology and global climate change (p. 3, l. 63-70).

Reviewer comment – revision 3

Thank you, however, please remember to reference every statement made.

Item 2

Reviewer comment – revision 1

Referring to comment 1.3 by reviewer 1, more details is need on what it is that makes psychological research methods distinct?

Author response

We have added a more detailed explanation: “A separate investigation of psychological PLSs is essential for at least two reasons: first, the distinctness of psychological research: as the subject is to examine the mind, the constructs of interest are often not directly observable, but must be inferred or operationalized by manipulating proxy variables. Furthermore, studies often rely on correlation designs and context-sensitive interpretations. Thus, psychology is a very nuanced and complex field with a growing complexity of research methods. Since it is a relatively new field, methods and subdisciplines are evolving rapidly. And second, the specificity of psychological jargon, in that the technical terms seemingly overlap with plain language.” (p.6, l. 134-142)

Reviewer comment – revision 2

I appreciate this detailed response. A minor note – authors state that there are two reasons, however I have located four separate points, all valid in their own right. For example:

1. the distinctness of psychological research: as the subject is to examine the mind, the constructs of interest are often not directly observable, but must be inferred or operationalized by manipulating proxy variables.

2. Furthermore, studies often rely on correlation designs and context-sensitive interpretations

3. psychology is a very nuanced and complex field with a growing complexity of research methods. Since it is a relatively new field, methods and subdisciplines are evolving rapidly.

4. the specificity of psychological jargon, in that the technical terms seemingly overlap with plain language

Re-writing this paragraph for clarity would be helpful. It should also be referenced throughout.

Authors' response

Thank you for your feedback. We re-wrote the paragraph to improve clarity and added references (p. 5, l. 127-142).

Reviewer comment – revision 3

I note the revisions to this section, however it still lacks clarity and contains jargon or technical terms not explained to the reader. The introduction of the Big Five Model of personality is poorly explained, as is the “recent phenomenon that has come to light is the overuse of psychological technical terms in everyday language, which results in the expansion of their meaning”. Examples of these technical terms would strengthen the argument better than the introduction of the Big Five Model. It should be noted however, that technical terms and jargon is not unique to the field of psychology, as the authors seem to imply in lines 133-134.

Authors' response

As recommended by the reviewer, we added examples of technical terms to strengthen our arguments. (p. 6, l. 138-148) as well as lines 133-135.

Item 3

Reviewer comment – revision 1

It would be useful to understand the participant selection process more fully to ensure there was no bias i.e., how were participants selected from the pool of respondents i.e., how they aimed “to maximize variation of socio-demographic characteristics (age, sex, education level) in the sample”. Were group numbers for each cohort determined prior to recruitment?

Author response

We describe the recruitment and selection process more clearly now:

1. We have stated that the number of groups was determined before recruitment started: “We aimed to conduct two focus group interviews with laypeople, one focus group interview with science communicators, and one with psychologists working as counsellor and/or psychotherapists (“psychological practitioners”) (p.8, l. 202-04)

2. We have provided more details on the recruitment process in S5 table.

3. We have explained why we had to select participants who fulfilled the inclusion criteria: “For laypeople, expressions of interest exceeded the number of available spots, so one researcher (CB) selected participants with the aim of maximizing variation in socio-demographic characteristics (age, sex, education level) within this group”. (p.10, l.240-43)

Reviewer comment – revision 2

1. Addressed with thanks.

2. Thank for the additional detail regarding recruitment. I note on p9 authors state “The detailed recruitment strategy is reported in S5 Table.” I find the level of information in this table is lacking in detail. I do not understand why this information was removed from the Methods section and put in a supplementary table. It could easily be summarised in a few sentences in the Methods section.

3. Please refer to my original question, in which I asked ‘how’, not ‘why’ were participants selected from the pool or respondents i.e., i.e., how did authors aim “to maximize variation of socio-demographic characteristics (age, sex, education level) in the sample”

Authors' response

Thank you for your comment. We included the information of S5 table in the text again (p. 9, l. 224-233).

Regarding the selection process, we describe the process more clearly now: “After voluntarily expressing their interest to participate via e-mail, potential participants received an informational letter on the procedure and conditions, and a consent form along with a short questionnaire on inclusion criteria and sociodemographic characteristics (age, gender, education level). Inclusion criteria were sufficient German language skills, legal age, interest in psychology and no degree in psychology (only laypeople) or a master’s degree in psychology and being psychological counsellors and/or psychological therapists (only psychological practitioners) or professional involvement with science communication including psychological topics (only science communicators). No other characteristics were collected that could indicate any attitudes toward PLSs. For laypeople, the number of submitted consent forms exceeded the number of available spots. While we did not employ stratified sampling or use a recruitment panel, we aimed to maximize variation in socio-demographic characteristics (i.e. a balanced gender distribution, different age decades and inclusion of the few participants with lower education levels) by purposively selecting individuals to achieve a heterogeneous sample. For psychological practitioners or science communicators, the number of submitted consent forms did not exceed the number of available spots.” (p. 9, l. 235-251)

Reviewer comment – revision 3

Thank you for response, however I note that the inclusion of the addition detail regarding the recruitment process (p9, lines 235-251) appeared in revision 2 and as such were not written in response to my repeated requests for information to clarify ‘how’, not ‘why’ were participants selected from the pool or respondents i.e., i.e., how did authors aim “to maximize variation of socio-demographic characteristics (age, sex, education level) in the sample”.

Additionally, I recommend clarifying the following:

• What is meant by “sufficient German language skills”

• What is meant by “legal age” as this may vary between countries

This paragraph is quite difficult to follow. I is long and the section relating to an interest in psychology, degree in psychology, science communication etc is not clear.

I recommend using formatting the inclusion criteria as a list instead of a paragraph.

Authors' response

Thank you for your comment. We describe the recruitment process in the text now (p. 10, l. 257-266): “For laypeople, the number of submitted consent forms exceeded the number of available spots. While we did not employ stratified sampling or use a recruitment panel, we aimed to optimise variation in socio-demographic characteristics by purposively selecting individuals to achieve a heterogeneous sample. Therefore, we included the few individuals with lower education levels and ensured a balanced gender distribution as well as representation of all age decades. For psychological practitioners or science communicators, the number of submitted consent forms did not exceed the number of available spots.” Since we did not use software to calculate the variation precisely but instead selected participants based on an overview of characteristics, we use “optimise” instead of “maximize”. Moreover, we have changed “sufficient German language skills” into “native-level German language proficiency (self-reported)” and clarified “legal age” (≥ 18 years), p. 10, l. 244-46.

Item 4

Reviewer comment – revision 1

Table 1 – Participant characteristics. I suggest the authors remember that this paper is likely to appeal to an international audience, so providing context for education levels would be helpful.

Author response

To provide context without going beyond the scope, we have classified educational levels now as low, moderate and high, and summarised the categories qualification for university and university degree in Table 1.

Reviewer comment – revision 2

I appreciate the effort the reclassify educational qualifications, however, I do not think that the using low, moderate and high is a good choice in terms of language. It could be alienating for some readers.

I also note that Table 1 appears to have moved to the Methods section. Since it reports study findings, it should be in the Results section.

Authors' response

Thank you for your comment. We partially reclassified the education levels according to the UNESCO International Standard Classification of Education. We have also rearranged Table 1 in the Results section and included a new short paragraph titled “Participants’ characteristics” (p. 13, l. 310-14).

Reviewer comment – revision 3

I note the addition of a short paragraph on p13 titled “Participant characteristics”. The purpose of this should be to summarise the key/notable findings in Table 1, and include results for age such as mean, range and median. Table 1, should only report frequency for the characteristic of ‘age’ for each age range, not median, range and mean.

Autors’ response

We have now provided additional information on participants’ characteristics and included the frequency of each age range in Table 1, instead of the mean, range and median (p.13, l. 323-326).

Item 5

Reviewer comment – revision 1

Line 192 – I would have liked to review the three PLSs that were part of the interviews, however S3 file was only provided in German. I am fluent in English only.

Author response

We have now provided an English translation of the three PLS in S4 File. Please note that this is a translated version of the original PLSs provided for informational purposes only. Due to the nature of plain language summaries, the exact content and complexity (compared to scientific language) of the original cannot be fully reproduced in translation.

Reviewer comment – revision 2

Thank you for providing the PLSs translated into English as requested. I would have also liked to see the guidelines in English, but acknowledge that I did not request these.

Authors' response

We appreciate the reviewer’s interest in the guideline. We fully agree that making it available in English would be valuable. However, due to time constraints, we are unable to provide an English version at this stage. We plan to translate the guideline and make it publicly available next year.

Reviewer comment – revision 3

Noted with thanks.

Item 6

Reviewer comment - original

Line 707 – This statement is unsupported - “For example, it is possible that people with a lower level of education did not feel motivated to participate in the study”.

Author response - original

We reformulated this sentence so that it is clear that this is only one possible hypothetical explanation: “For example, there is a possibility that it is mainly higher educated people who are interested in taking part in psychological studies on psychological plain language summaries – and that it is also these people who are more likely to read psychological PLSs.” (p.37, l.758)

Reviewer comment – follow up 1

I appreciate that this statement has been reworded, however it still needed to be cited with supporting evidence.

Authors' response follow up 1

We added a citation of a study investigating education bias to strengthen our reasoning (p. 36, l. 734-737).

Reviewer comment - revision 3

Although there have clearly been additional references added as the total number of references has increased from N=31 in revision 2 to N=36 in revision 3, there are two issues with the use of references in this section (p36, lines 734-737).

1. There are zero references used in any of these lines to support the statements made by the authors The following sentences require references:

“However, the aim of PLSs is to help people understanding scientific research rather than supporting individual decisions with a complete presentation of options and implications. Psychological practitioners, however, considered the PLSs to be valuable informational material to be shared with interested patients.”

2. The references cited at the end of this paragraph i.e., [7, 36] were not newly added to this version, appearing in revision 2.

I see no evidence that authors have provided a reference in this section as they stated in their response.

Authors’ response

We added reference [35] in revision 2 (p.36, l.743-44). We now added reference Stoll et al. [8] for the cited statement “However, the aim of PLSs is to help people understanding scientific research rather than supporting individual decisions with a complete presentation of options and implications.” (p.36, l.762-764). The other statement (“Psychological practitioners, however, considered the PLSs to be valuable informational material to be shared with interested patients.”) is a result of the current study and therefore not cited. We now begin the sentence with “As this present focus group shows”, so that this becomes clearer (p.36, l.764).

Item 7

Reviewer comment – revision 2

When introducing the acronym (PLSs) for plain language summaries, all letters should be lower case, unless used at the start of a sentence i.e., plain language summaries (PLSs). This is in keeping with the established convention of other published research in the field, including the cited work by Stoll et al., 2022.

Authors' response

The term is written now in lowercase when we introduce it (p. 4, l. 80).

Reviewer comment – revision 3

Noted.

---

## [Editor Report · Decision Letter 4]

9 Feb 2026

"A very first clue on the subject": A focus group study on users' perspectives on German plain language summaries of psychological meta-analyses

PONE-D-24-22614R4

Dear Dr. Breuer,

We’re pleased to inform you that your manuscript has been judged scientifically suitable for publication and will be formally accepted for publication once it meets all outstanding technical requirements.

Kind regards,

Muhammad Zammad Aslam, Ph.D.

Academic Editor

PLOS One
---

## [Editor Report · Acceptance letter]

PONE-D-24-22614R4

PLOS One

Dear Dr. Breuer,

I'm pleased to inform you that your manuscript has been deemed suitable for publication in PLOS One. Congratulations! Your manuscript is now being handed over to our production team.

Kind regards,

on behalf of

Dr. Muhammad Zammad Aslam

Academic Editor

PLOS One